# Different sustained and induced alpha oscillations emerge in the human auditory cortex during sound processing

Víctor J. López-Madrona [1,2] ✉, Agnès Trébuchon[3,4], Christian G. Bénar[2], Daniele Schön [2] & Benjamin Morillon [2]

Alpha oscillations in the auditory cortex have been associated with attention and the suppression of irrelevant information. However, their anatomical organization and interaction with other neural processes remain unclear. Do alpha oscillations function as a local mechanism within most neural sources to regulate their internal excitation/inhibition balance, or do they belong to separated inhibitory sources gating information across the auditory network? To address this question, we acquired intracerebral electrophysiological recordings from epilepsy patients during rest and tones listening. Thanks to independent component analysis, we disentangled the different neural sources and labeled them as "oscillatory" if they presented strong alpha oscillations at rest, and/or "evoked" if they displayed a significant evoked response to the stimulation. Our results show that 1) sources are condition-specific and segregated in the auditory cortex, 2) both sources have a high-gamma response followed by an induced alpha suppression, 3) only oscillatory sources present a sustained alpha suppression during all the stimulation period. We hypothesize that there are two different alpha oscillations in the auditory cortex: an induced bottom-up response indicating a selective engagement of the primary cortex to process the stimuli, and a sustained suppression reflecting a general disinhibited state of the network to process sensory information.

The brain is always active, with brain oscillations present even in the absence of any particular stimulus[1,2]. Alpha ( ~ 10 Hz) oscillations are widely distributed across the cortex and are believed to play a key role in attention, controlling the excitation/inhibition balance and inhibiting the regions nonrelated to the task to gate the information flow between relevant distributed networks[3–6]. An alpha-like rhythm has been described in the auditory cortex[7–9]. It has a lower frequency than the visual or sensorimotor alpha (6-10 Hz[10–12]), but has been suggested to play an equivalent role, controlling the balance between excitation and inhibition in the auditory cortex[13]. However, the anatomical and functional organization of these oscillations is unclear.

During speech processing, alpha activity is suppressed in regions responding to the stimulus, suggesting a local bottom-up disinhibition to favor information processing[14,15]. On the other hand, alpha activity has also been associated to top-down anticipatory processes[14], which may require sources located in higher level areas to control the excitation/inhibition balance in other regions. In this work, we want to explore the nature and

function of alpha sources in the auditory cortex. What are the neural sources of alpha activity, its response profile during auditory stimulation and their relationship with the evoked neural sources processing the sensory input.

Most previous studies were based on non-invasive recordings, which lack the spatial resolution to identify and localize the neural generators of different oscillatory dynamics. Intracerebral recordings in the form of stereotaxic electroencephalography (SEEG) are an excellent approach to characterize the dynamics of the auditory cortex at a millisecond time scale and fine spatial specificity[16]. However, although SEEG electrodes are located directly in the region of interest, each contact may still record the activity from multiple brain sources due to volume conduction. Independent component analysis (ICA) is a methodology that aims to separate the time-courses of the different current generators contributing to the recorded field potentials[17–19]. It has been extensively used in non-invasive recordings, both to remove artifacts such as cardiac activity or blinks[20] and to retrieve neural sources[21–23].

In intracerebral recordings, ICA has the potential to outperform traditional montages[17,18]. In referential montages, each contact records the

[1]Institute of Language, Communication, and the Brain, Aix-Marseille Univ, Marseille, France. [2]Aix-Marseille Univ, INSERM, INS, Inst Neurosci Syst, Marseille, France. [3]APHM, Timone Hospital, Epileptology and cerebral rhythmology, Marseille, 13005, France. [4]APHM, Timone Hospital, Functional and stereotactic neurosurgery, Marseille, 13005, France. ✉e-mail: victor.LOPEZ-MADRONA@univ-amu.fr

**Fig. 1 | Separation of oscillatory and evoked sources with ICA. a** Cerebral MRI scan (3D T1-weighted) – cross section with reconstruction of the SEEG electrode for patient 9. The location of each contact is represented with white rectangles. **b** Example of monopolar recordings during rest. High amplitude oscillations can be appreciated in the superior channels (H'9 and H'10). **c** Averaged AEP during presentation of pure tones at each contact. The highest response in amplitude is observed in channel H'4. **d** Spatial profile of three SEEG-ICs across the electrode, representing their contribution to each contact. The purple and blue components have clear peaks in the profile, suggesting a local origin of the sources around these contacts. The grey component contributed almost equally to all the contacts, and hence reflect a remote source. **e** SEEG-IC traces during the same time period as (**b**). The oscillations visible in the raw SEEG were captured by the purple component, which was labeled as oscillatory source. **f** Averaged AEP of each SEEG-IC (solid lines) and a single trial (dashed lines). Only the blue component has a significant response and was labeled as evoked source.

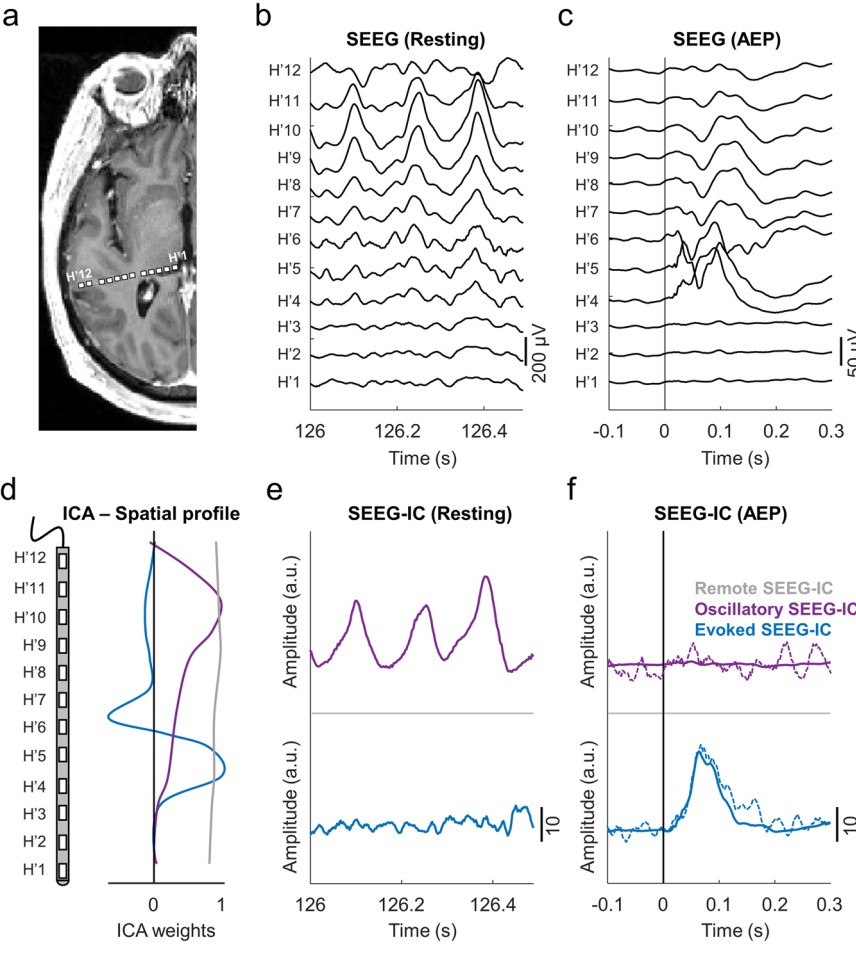

activity of both local and remote sources that may be located far away[24]. One approach is to identify and remove the distant sources, whether it represents the electrical reference[25,26], or other neural sources[27]. Moreover, rather than discarding this activity, it is possible to localize and analyze it, similar to the inverse problem in non-invasive recordings[24,28]. Bipolar montages are commonly used to measure local currents in a given location, but they may not recover the correct time-courses of local sources[27,29,30]. For example, if two sources are located close to the same SEEG contact, the bipolar montage would not be able to separate them[31]. Therefore, ICA can be used to separate the multiple sources of alpha activity in the auditory cortex.

In this work we performed SEEG recordings from the human auditory cortex to track the activity of the neural sources during two conditions: rest and pure tone stimulation. With ICA, we identified the main sources of alpha oscillations at rest ("oscillatory sources") and those with a significant auditory evoked potential (AEP, "evoked sources"). First, we compared whether the sources responding to the stimulus were also those with highest alpha at rest (i.e., whether they were the same neural source or not). Second, we compared the power spectrum across both conditions to characterize sustained changes of alpha power. Finally, we analyzed the time-frequency response of the sources during pure-tone stimulation for a fine-grained exploration of stimulus induced alpha modulations.

## Results
### Separation of resting and auditory sources with ICA
We recorded the SEEG activity from the auditory cortex (Fig. 1a) of 18 epileptic patients during rest and during the presentation of pure tones (Fig. 1b, c). We selected a total of 23 electrodes (5 patients having a bilateral implantation), each with multiple channels (between 8 and 18, total channels: 305). For each electrode, the time courses of both conditions were concatenated, and the ICA source separation method was applied to

segregate the recordings into the main neural sources contributing to the SEEG activity (Fig. 1d). A single mixing matrix (i.e., a set of spatial profiles) was obtained for each electrode, allowing traceability of the same sources across both conditions (Fig. 1e, f). After removing the components associated to remote sources (Fig. 1d, gray component), a total of 284 SEEG independent components (SEEG-ICs) were selected across electrodes. We then classified each SEEG-IC as oscillatory source, evoked source, both oscillatory and evoked source or nonrelated component (see methods). A total of 58 SEEG-ICs presented a significant auditory evoked potential (AEP; $N = 23/23$ electrodes) and were, therefore, labeled as evoked, while 71 SEEG-ICs were labeled as oscillatory ($N = 21/23$ electrodes). Interestingly, from all the SEEG-ICs, only 11 were labelled as both oscillatory and evoked, suggesting that the neural sources with the main oscillatory activity in rest were not the sources processing the stimulus (they did not have a significant AEP).

The explained variance of the SEEG-ICs (i.e., contribution of the component to all the SEEG recordings) were different for both sources. Each oscillatory source explained between 0.4% and 63% (mean 10.5%) of the total activity of the data, while evoked sources captured between 0.4% and 27.5% (mean 6.4%) of the variance, suggesting that spontaneous activity was predominant in our recordings.

To further check that the obtained sources were not driven solely by one of the two conditions, we repeated the ICA procedure on each condition separately and computed the correlation between the previous SEEG-ICs (with both conditions concatenated) and the new time courses (computed separately for both conditions; see methods). The oscillatory components were quite stable across analyses, and the same components were retrieved when ICA was computed only on the rest condition (averaged correlation of SEEG-ICs ± standard deviation, s.d.: 0.84 ± 0.13) or only during the stimulation condition (0.84 ± 0.14). Results were similar for the evoked

**Fig. 2 | Location of oscillatory and evoked SEEG-ICs. a** Comparison of the location of oscillatory and evoked SEEG-ICs along the lateral-medial axis, grouping both hemispheres (i.e., absolute value of the x-axis in MNI space; **p < 0.01, t-test). The central mark of the box indicates the median, and the bottom and top edges of the box indicate the 25th and 75th percentiles. The whiskers extend to the most extreme data points not considered outliers, and the outliers are plotted individually with black crosses. **b** Position of the SEEG-ICs on a 3D surface of the temporal lobe. The location of each SEEG-IC in a common MNI space is measured as the contact with maximal contribution in the spatial profile.

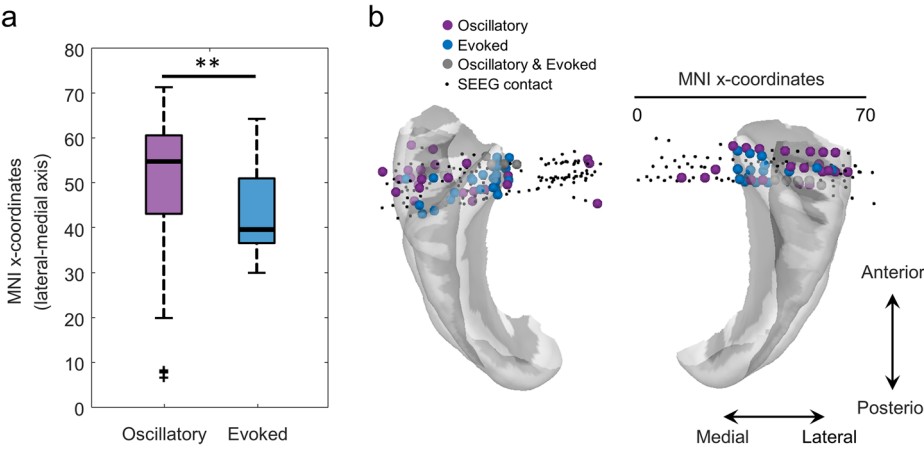

sources, although it was more difficult to retrieve them when analyzing only the rest condition (averaged correlation of SEEG-ICs: 0.76 ± 0.13), compared to the pure tone condition (0.81 ± 0.15). Therefore, the two types of sources were present at both rest and during stimulation, rather than being active only in one condition and completely silent during the other.

The location of the SEEG-ICs differed for both types of sources (average of absolute lateral-medial axis locations in MNI space ± s.d.: 50.39 ± 11.65 and 43.49 ± 8.89 for oscillatory and evoked SEEG-ICs; p = 0.0024, t-test, t = 3.09, df = 125; Fig. 2a), with evoked sources located in medial areas while oscillatory sources were more lateral. The contacts with maximal contribution of the oscillatory SEEG-ICs were distributed along the lateral-medial axis, in contrast to the evoked sources, which were clustered in more medial areas (Fig. 2b). The oscillatory sources occupied a larger area (i.e., they were more distributed) than the evoked sources, with their location presenting a higher standard deviation across SEEC-ICs (s.d. of the absolute lateral-medial axis locations in MNI space: 11.65 and 8.89 for oscillatory and evoked SEEG-ICs, p = 0.03, permutation test).

Regarding the anatomical region of each SEEG-IC (Table 1), the oscillatory sources were predominately located in the Superior Temporal Sulcus (STS) and the areas 22 and 41/42 of the Superior Temporal Gyrus. The evoked sources were mainly in the STG (areas TE1.0 and TE1.2; primary auditory cortex[32]) and the insular gyrus. None of the evoked SEEG-ICs were identified in the STS.

## Resting oscillations are attenuated during pure tone stimulation

To further characterize the dynamics of the oscillatory SEEG-ICs, we analyzed their relative power spectrum, by removing the aperiodic activity following the *fooof* approach (Donoghue et al.[33]; Fig. 3a; see methods). All the SEEG-ICs labeled as both oscillatory and evoked were removed from the

analysis (resulting in N = 60 purely oscillatory ICs). For each purely oscillatory SEEG-IC, we measured the frequency and relative power of its main oscillatory activity and the exponent and offset of the aperiodic component of its power spectrum (Fig. 3b). The frequencies of the oscillations were relatively narrow, ranging between 5 and 10 Hz (frequency peak: 7.9 ± 1.04 Hz and 8.1 ± 1.4 Hz, mean ± s.d. for rest and pure tone conditions), with no differences between rest and pure tone stimulation (paired t-test, p > 0.15, t = −1.48, df = 59, CI = −0.58, 0.09, effect size = −0.25; Fig. 3c). Since frequency resolution can impact the quality of the fit in the FOOOF algorithm, we repeated the analysis with a higher resolution (0.05 Hz; original resolution = 0.25 Hz; see Methods) with equivalent results (frequency peak: 7.9 ± 1.06 Hz and 7.9 ± 1.09 Hz, mean ± s.d. for rest and pure tone conditions; paired t-test, p > 0.7, t = −0.38, df=59, CI = −0.08, 0.06, effect size = −0.01). However, the relative power of the oscillatory activity strongly decreased during the pure tone condition (1.04 ± 0.23 arbitrary units, a.u., and 0.67 ± 0.32 a.u. for oscillatory SEEG-ICs during rest and stimulation; paired t-test, p < 0.001, t = 8.23, df = 59, CI = 0.28, 0.46, effect size = 0.37; Fig. 3d), suggesting an interruption of the cortical rhythmicity during listening. The aperiodic features also presented slight differences, with higher values during rest (exponent: 1.73 ± 0.48 a.u. and 1.65 ± 0.41 a.u. for oscillatory SEEG-ICs during rest and stimulation; paired t-test, p = 0.0095, t = 2.68, df=59, CI = 0.02, 0.14, effect size = 0.08; offset: −0.27 ± 0.40 a.u. and −0.43 ± 0.45 a.u.; paired t-test, p < 0.001, t = 3.54, df = 59, CI = 0.07, 0.25, effect size=0.16), indicating a steeper slope during the resting condition.

Then, we tested whether the dynamics of the oscillatory SEEG-ICs were dependent of their location within the auditory cortex, computing the correlation between the spectral features of the sources and their anatomical location in MNI coordinates along the lateral-medial axis (same direction as the electrode). Although the oscillatory sources covered a relatively broad section in this axis, we could not identify any gradient between the analyzed features and the location (Supplementary Fig. 2), suggesting that oscillatory dynamics are independent of the depth of the neural source.

Although the oscillatory SEEG-ICs did not have a significant AEP during pure tone stimulation, they presented significant responses in the time-frequency domain (Fig. 3e). An initial response was elicited at low gamma frequencies (20–40 Hz; 25–100 ms), followed by a high-gamma activation (80–120 Hz) that lasted from 50 to 150 ms after stimulus onset (p < 0.05, corrected with FDR). This increase of activity was followed by a suppression at low frequencies between 5 and 30 Hz that started at 200 ms. To test that a single patient with several SEEG-ICs was not driving the results, we repeated the analysis by averaging all the oscillatory sources per subject (number of observations equal to the number of patients with an oscillatory source), obtaining the same time-frequency pattern (Supplementary Fig. 3).

## Table 1 | Anatomical location of each SEEG-IC

| Region | # Oscillatory SEEG-ICs | # Evoked SEEG-ICs |
|---|---|---|
| STG (A41/A42) | 13 | 8 |
| STG (A22c) | 7 | 3 |
| STG (A22r) | 4 | 3 |
| STG (TE1.0 and TE1.2) | 5 | 11 |
| STS (cpSTS) | 6 | 0 |
| STS (rpSTS) | 6 | 0 |
| Insular gyrus | 3 | 13 |
| Other | 16 | 9 |
| **Total:** | **60** | **47** |

*STG* Superior Temporal Gyrus, *A22c* caudal area 22, *A22r* rostral area 22, *STS* Superior Temporal Sulcus, *cpSTS* caudoposterior STS, *rp* rostroposterior STS.

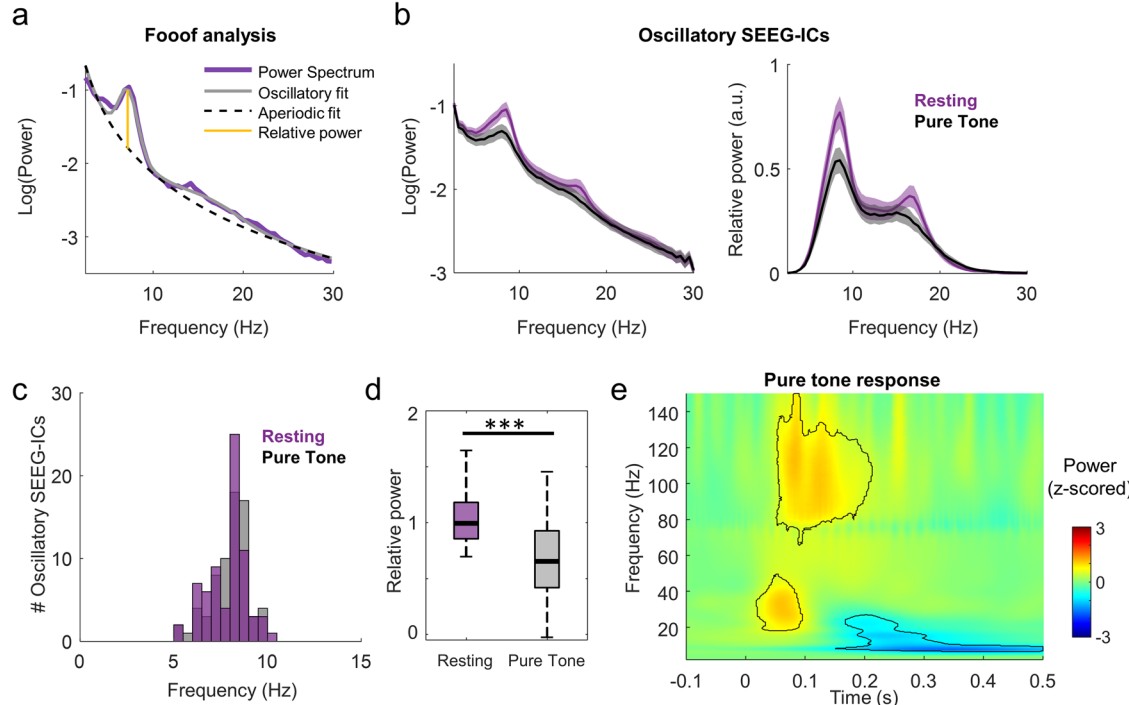

**Fig. 3 | Dynamics of oscillatory SEEG-ICs during rest and pure tone stimulation.** **a** Scheme of the *fooof* approach. The power spectrum of a SEEG-ICs is modeled by the combination of the oscillatory sources and an aperiodic activity. The relative power of an oscillatory rhythm is then measured as the difference between both sources (red line). **b** On the left, averaged power spectrum across oscillatory SEEG-ICs during rest and pure tone stimulation (N = 60, mean ± s.e.m.). On the right, relative power of the oscillatory SEEG-ICs during both conditions, measured as the difference between the oscillatory and the aperiodic fit. **c** Histogram of main frequency peaks of the oscillatory sources during rest and pure tone stimulation. **d** Comparison of the relative maximal power of the oscillatory SEEG-ICs obtained with the *fooof* approach between conditions (***$p < 0.001$). **e** Time-frequency response of the average oscillatory SEEG-ICs during pure tone stimulation. Framed areas represent clusters of significant modulation of activity compared to baseline ($-300 -200$ ms; $p < 0.01$, surrogate analysis).

To ensure that the results were not driven by our selection criteria of oscillatory sources (25% of SEEG-ICs with highest oscillatory power), we repeated the analysis selecting the 10% and 50% of SEEG-ICs with highest power. We found similar significant differences in relative power across conditions and similar time-frequency responses to pure tones in both cases.

### Medial sources are activated earlier than lateral sources during pure tone stimulation

We then characterized the dynamics of the evoked SEEG-ICs by analyzing the latencies of their evoked potentials (Fig. 4a). All the SEEG-ICs labeled as both oscillatory and evoked were removed from the analysis (resulting in N = 47 purely evoked ICs). There were a large variety of responses, scaling from 15 to 80 ms after stimulus onset and including positive and negative patterns. We focused on the earliest latency of each SEEG-IC as an indicator of its activation time (Fig. 4a). This response correlated with the SEEG-IC location in the lateral-medial axis (Fig. 4b, Pearson correlation, $R = 0.336$, $p = 0.021$, $t = 2.396$, df = 45). The first SEEG-ICs responding to the stimulus were in the deeper areas of the auditory cortex, followed by a sequential activation of SEEG-IC sources in the lateral direction (Fig. 4c).

As for the oscillatory SEEG-ICs, we analyzed the spectral content of the evoked SEEG-ICs (Fig. 4d). To control for the contribution of the AEPs to the power spectrum, we measured the power spectrum either on the raw signal (Fig. 4d, Pure Tone) or after removing the averaged evoked response (AEP; i.e., the phase-locked activity) from each trial. We then estimated their relative power with the *fooof* approach (Fig. 4d, Pure Tone no AEP; see methods). First, the relative alpha power in these evoked sources was minimal compared to the oscillatory SEEG-ICs (evoked: 0.20 ± 0.20 a.u.; oscillatory: 1.04 ± 0.23; in the Resting condition). Then, the relative alpha power was apparently higher during rest than pure tone (0.20 ± 0.20 a.u. and 0.17 ± 0.15 a.u. for evoked SEEG-ICs during rest and stimulation; paired t-test, $p = 0.009$, t = 2.75, df = 46, CI = −0.014, 0.094, effect size = 0.05; Fig. 4e,

left). However, this effect was induced by the AEP, as it was not present after removing the phase-locked activity (Resting vs. Pure Tone no AEP: 0.20 ± 0.20 a.u. and 0.19 ± 0.17 a.u. for evoked SEEG-ICs during rest and stimulation without AEP; paired t-test, $p = 0.69$, t = 0.397, df = 46, CI = −0.05, 0.07, effect size=0.01). This indicates that in these evoked neural sources, alpha oscillatory activity does not vary between rest and stimulation.

The aperiodic features were also strongly driven by the AEP (Fig. 4e). Both the exponent and the offset were apparently higher during pure tone stimulation than rest (exponent: 1.24 ± 0.38 a.u. and 1.54 ± 0.42 a.u. for evoked SEEG-ICs during rest and stimulation; paired t-test, $p < 0.001$, t = −5.03, df=46, CI = −0.30, −0.13, effect size = −0.21; offset: −0.59 ± 0.37 a.u. and −0.20 ± 0.39 a.u., paired t-test, $p < 0.001$, t = −5.16, df=46, CI = −0.44, −0.19, effect size = -0.32). However, after removing the effect of the evoked response (AEP), we observed the opposite effect, similar to what was found for the oscillatory sources, with significantly higher values during rest, indicating a steeper slope in the resting condition (exponent: 1.24 ± 0.38 a.u. and 1.05 ± 0.29 a.u. for evoked SEEG-ICs during rest and stimulation without AEP; paired t-test, $p < 0.001$, $t = 5.20$, df = 46, CI = 0.11, 0.26, effect size=0.18; offset: −0.59 ± 0.37 a.u. and −0.83 ± 0.29 a.u., paired t-test, $p < 0.001$, t = 5.18, df = 46, CI = 0.15, 0.34, effect size=0.24). Of note, no correlation was found between the location of the sources in the lateral-medial axis and the main features of the power spectrum during the stimuli: relative power, exponent and offset of the aperiodic activity (Supplementary Fig. 2).

### Induced alpha responses are equivalent between oscillatory and evoked sources

The time-frequency response of the evoked SEEG-ICs was also driven by the AEP (Fig. 5). The waveform of the auditory response had a ~ 10 Hz pattern and most of the activity at low frequencies was strongly phase-locked with

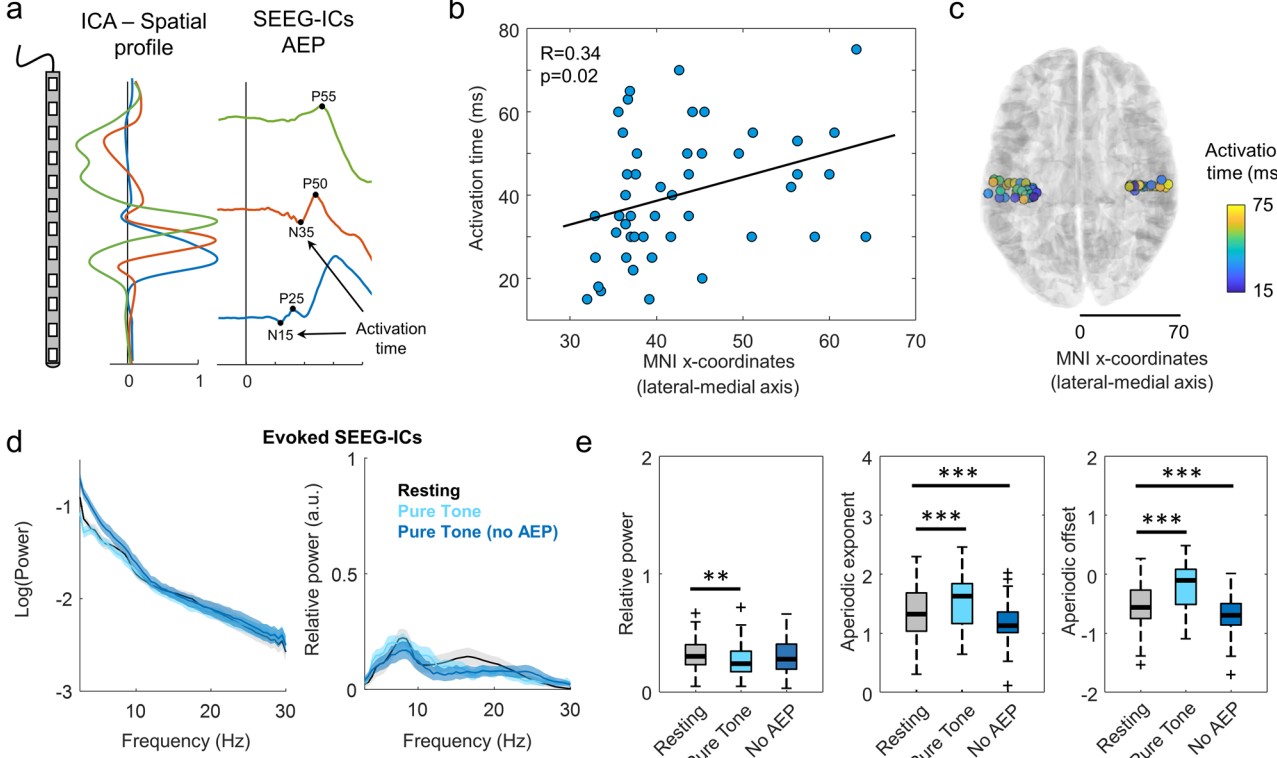

**Fig. 4 | Dynamics of evoked SEEG-ICs during rest and pure tone stimulation.**
**a** Example of evoked SEEG-ICs in patient 4. The source with the most medial topography (i.e., peak of the spatial profile in deeper contacts) has a faster activation time (blue trace). **b** Pearson correlation (black line) between the location of the evoked SEEG-ICs in the lateral-medial axis and the activation time (blue circles). **c** Location of the evoked SEEG-ICs in a common MNI space. The color of each circle indicates the activation time of the source located in that location. **d** On the left,

averaged power spectrum across evoked SEEG-ICs during rest and pure tone stimulation, either including or removing the contribution of the AEP to the time-course ($N = 47$, mean ± s.e.m.). On the right, relative power of the evoked SEEG-ICs for each condition. **e** Comparison of three features extracted with the fooof approach for each condition: relative power (left panel), exponent of the aperiodic component (middle) and offset (right; **/***, $p$-val < 0.01/0.001, respectively; paired t-test).

the stimulus (Fig. 5a, right panel). The AEP also created a chimney effect in the time-frequency spectrum, close to stimulus onset. Therefore, neural activity in the entire frequency range was affected by the AEP and cannot be directly associated with changes in the oscillatory dynamics. To mitigate this effect, we also analyzed the response after removing the averaged evoked response (AEP; i.e., the phase-locked activity) from each trial (see Methods). Without the AEP contribution, the time-frequency map of the evoked sources had some similarities with the oscillatory sources (Fig. 5b vs. Fig. 3e). There was an early activation (here at very low frequencies; ~10 Hz) with a strong high-gamma response that started 25 ms after stimulus onset and lasted 200 ms, followed by a suppression at low frequencies (between 5 and 30 Hz), starting at 125 ms in these sources ($p < 0.05$, corrected with FDR). We repeated the analysis by averaging all the evoked SEEG-ICs per subject (number of observations equal to the number of patients with an evoked source), obtaining the same time-frequency pattern (Supplementary Fig. 3).

As the frequency of the AEP waveform overlapped with the alpha range, we could not dissociate whether the phase-locked activity was also contributed by a phase resetting of the ongoing oscillations. To better explore this scenario and knowing that the phase of ongoing oscillations in auditory cortex likely modulates its responsiveness to incoming stimuli[34–36], we measured how the instantaneous phase of alpha at stimulus onset influenced three different features of the response: amplitude of the AEP, high-gamma increase and alpha decrease (see Methods). We computed the weighted ITPC for each SEEG-IC, estimating their significance at the single level (permutation test). In only 2/60 oscillatory SEEG-ICs and 3/47 evoked SEEG-ICs, the phase of alpha oscillations influenced the intensity of the power response, either in the alpha or high-gamma range (Supplementary Fig. 4). This indicates a marginal influence of alpha phase on the power response. For the amplitude of the AEP, 15/46 evoked sources did present a

significant link with the instantaneous phase of alpha, although this effect was not significant at the group level.

Then, we compared the differences in the time-frequency activation between oscillatory and evoked sources (unpaired t-tests, corrected with FDR; Fig. 6a). The high-gamma (80-120 Hz) response after stimulus onset (10-200 ms) was significantly higher in the evoked sources compared to the oscillatory sources ($p < 0.05$, corrected with FDR). The short-lived low frequency (< 15 Hz) early response was also significantly stronger in the evoked sources. Note that this response does not coincide in time with the alpha suppression identified in Figs. 3e and 5b (which is not significantly different across sources) and instead coincides in time with the evoked potential. The low-gamma response (~ 20-40 Hz) also identified between 100 and 200 ms after stimulus onset was significantly higher in the oscillatory sources.

Finally, we tested whether the activation time between oscillatory and evoked sources differ during pure tone stimulation. We measured the evoked-related power at high-gamma (80-120 Hz) frequencies (Fig. 6b) and measured the onset latency of this activity. Evoked sources responded significantly faster than oscillatory sources (68 ± 41 ms and 33 ± 24 ms for oscillatory and evoked SEEG-ICs, respectively; t-test, $p < 0.0001$, t = 4.71, df = 77, CI = 20, 50).

## Discussion
In this work, we have identified two distinct types of activity sources in the auditory cortex: one presenting high alpha-like oscillations (5-10 Hz) during rest that are suppressed during auditory stimulation (oscillatory sources); and a second one that is relatively silent at rest but presents a significant evoked response during pure-tone stimulation (evoked sources). Results show that these sources are spatially segregated: the evoked source

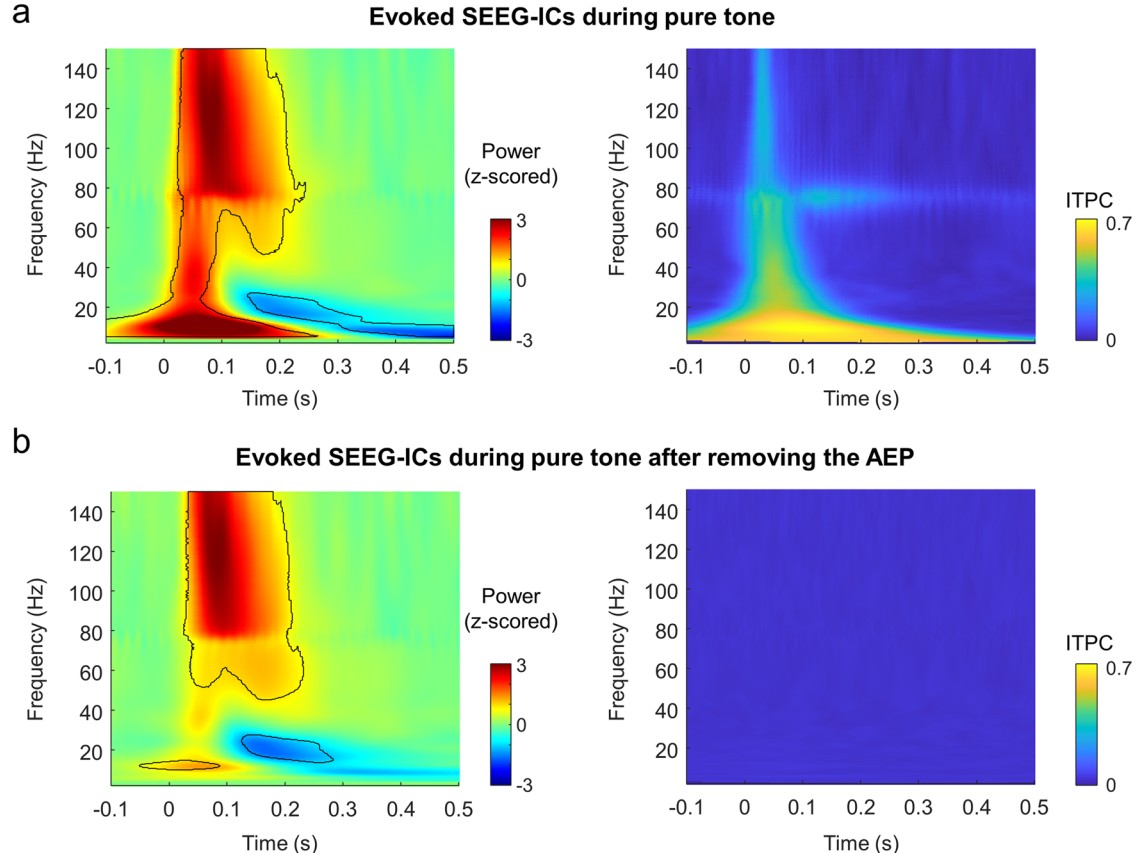

**Fig. 5 | Time-frequency and phase locking responses of evoked SEEG-ICs.**
**a** Averaged time-frequency response of evoked SEEG-ICs during pure tone stimulation (left panel). Framed areas represent clusters of significant modulation of activity compared to baseline (-300 -200 ms; $p < 0.01$, surrogate test). Inter-trial phase clustering (ITPC) representing the frequencies phase-locked with the stimulus (right panel). **b** Same analysis as in panel a but after removing the contribution of the averaged AEP to the evoked SEEG-IC time-courses. This process removes only the activity phase-locked with the stimulus.

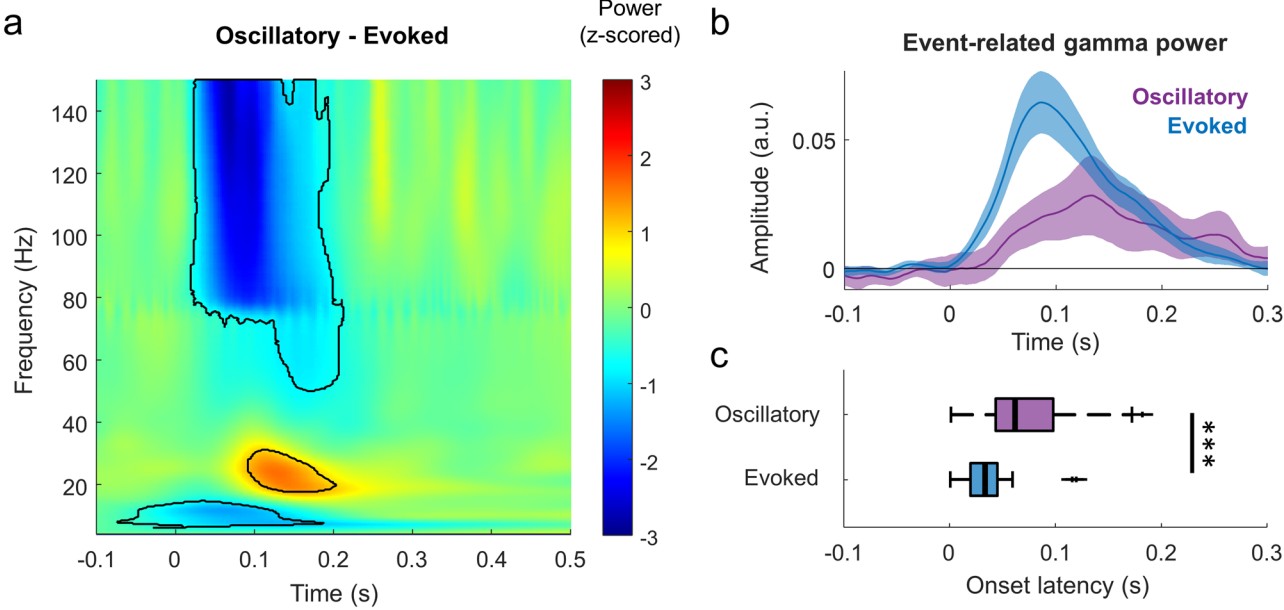

**Fig. 6 | Comparison of the time-frequency dynamics between oscillatory and evoked SEEG-ICs. a** Contrast between the time-frequency responses of oscillatory (Fig. 3e) minus evoked (Fig. 5b) sources. Delineated areas represent the clusters with significant difference ($p < 0.05$ corrected with FDR). **b** Averaged gamma power during pure tone stimulation of oscillatory and evoked SEEG-ICs (mean ± s.e.m.). **c** Onset latency of gamma activity for each source type. Evoked sources had a significant early activation than oscillatory SEEG-ICs suggesting a processing hierarchy.

dominates in medial, primary auditory areas, while the oscillatory source is more distributed including lateral higher-level auditory regions (Fig. 2). Both sources present a canonical time-frequency response to auditory stimulation, with a strong increase in high-gamma activity (80-120 Hz) followed by a decrease in the low-frequency range (5-30 Hz). A second decrease was also identified in the oscillatory sources in the alpha range (8 Hz). Unlike the transient effects observed in the time-frequency analysis, this decrease was not stimulus-locked but sustained throughout the entire pure tone stimulation task, compared to the resting condition (Fig. 3d). Finally, short-lived early responses were also observed, with a low-frequency ( ~ 10 Hz) increase of activity in the evoked sources, characteristic of evoked responses, and a low-gamma (20-40 Hz) increase of activity in the oscillatory sources.

Evoked sources were activated progressively from medial to lateral areas (Fig. 4b). This result, firstly reported in[16], is in line with the view of a hierarchical functional organization of the auditory cortex, where the information flows from primary to parabelt areas[37]. The onset latencies of the high-gamma response were also different between oscillatory and evoked sources (Fig. 6c), with a faster response on the evoked SEEG-ICs (i.e., with a significant AEP). However, the correlation between activation time and location in the lateral-medial axis was relatively low, and it included lateral sources that were activated earlier in time (Fig. 4b). Therefore, although the strict serial model seems the main route to process information, our results indicate the presence of multiple branches of communication between subcortical and (medial and lateral) cortical structures[38–40].

It has been suggested that the main role of alpha oscillations is to inhibit task-irrelevant regions, while a reduction of this rhythm would disinhibit the system to favor information processing[4,15,41]. In time-frequency maps, this is typically characterized as an increase of the high-gamma activity (information processing) followed by a suppression of the alpha activity (disinhibition) in regions engaged in the task[42]. In good agreement with previous studies[43,44], we found that the high-gamma activity was dissociated from the alpha response, being stronger in the evoked sources (Fig. 6a), mainly located in primary medial areas. There were no significant differences in alpha depression between evoked and oscillatory sources at the main timing of this effect (200-500 ms), suggesting that the amplitude of early high-gamma activity is not directly related to the amount of decrease of the alpha rhythm. Our results support the hypothesis of the stimulus-induced alpha suppression as a local bottom-up response to selectively engage the source to process the stimulus[45].

In addition to the induced alpha, we identified a sustained suppression of alpha oscillations during stimuli presentation (Fig. 3). This spontaneous rhythm has been described as an active suppression mechanism of cortical synchronization[13]. During the presentation of pure tones this rhythm is strongly suppressed, facilitating the processing of the new sensory input. Given the basic nature of the stimuli and the task (passively listening to pure tones), together with the huge effect on the alpha power, our results suggests that the main goal of this sustained alpha suppression in the auditory cortex is to facilitate the processing of every acoustic information, while smaller changes induced by the stimulus may represent a bottom-up process to attract attention[46] or to modulate the complexity of the stimuli[47].

It has been described a gradient in alpha power in the auditory cortex[43]. Compared to posteromedial structures, anterolateral areas have higher levels of alpha activity before auditory stimulation, followed by a stronger alpha suppression in response to sentences. Our work complements this result by describing two types of alpha suppression (sustained and induced). The evoked sources, mainly related to primary posteromedial structures, presented low alpha power during rest. The oscillatory sources, more distributed and covering lateral areas, were characterized by large alpha oscillations, corroborating the results reported before stimulus onset[43]. While the induced alpha suppression in response to pure tones was similar for both sources (Fig. 6a), the strong sustained alpha suppression, only present in oscillatory sources, may align with the identified gradient[43]. As the time interval between sentences in[43] was up to several seconds, it is possible that the auditory system had enough time to recover and reestablish the alpha power to a level similar to the resting condition.

It is worthy at this point to comment on the "oscillatory" nature of alpha. It has been suggested that the alpha cycle imposes the "opportunity windows" or "duty-cycle" of gamma activity[48]. However, our results suggest that the changes in alpha oscillations, defined as a sustained suppression in the oscillatory sources, may not be related with the induced alpha activity, given that both phenomena can be dissociated. In this scenario, we would have two systems: a sustained oscillatory alpha rhythm whose phase and amplitude are determinant of the inhibitory state of the network[49,50], and an alpha activity (not necessary an oscillation), whose suppression indexes the facilitation of information flow after stimulus onset. Whether both systems are related or not is a question that needs to be answered.

The suppression of the alpha activity found in our study cannot be accounted by changes in the $1/f^x$ component of the power spectrum as our methodology separated the oscillatory and aperiodic components[33]. Nevertheless, we also identified a decrease in the slope of the signal during pure tones in both sources[43]. This difference could be explained by a global increase in neural activity during the task[51] and may be linked to an increase in the excitation/inhibition balance of the network[52].

Two additional differences were identified when comparing the time-frequency map of the sources (Fig. 6a). A low-gamma (20–40 Hz) early response was present in the oscillatory sources, but not in the evoked ones. While this response may be related to the temporal duration of the pure tone stimulus (30 ms), its absence in the evoked sources (associated to primary areas and to stimulus response) is striking. One possibility is that the oscillatory sources have a higher sensitivity to stimuli with a low-gamma period[53–55], a key frequency for phonemic processing[56] that becomes left-lateralized in more associative (anterior STG; A4) regions[53,57]. A low-frequency short-lived early response was also specifically present in the evoked sources, which corresponds to the typical spectral signature of an AEP and would hence be caused by presence of vestiges of the AEP.

There are some limitations to consider in this study. One of them is the categorical differentiation between sources. While oscillatory and evoked sources can be identified and separated, the former exhibits an evoked response in the time-frequency plane (Fig. 3e) and the latter shows an alpha peak in the spectrum (Fig. 4d), suggesting that both phenomena are not completely dissociated. Therefore, the anatomical segregation in Fig. 2 could reflect a gradual change of alpha dominance. We did not observe any anatomical gradient of alpha power (Supplementary Fig. 2), supporting our hypothesis of two independent sources in the auditory cortex.

A second limitation of our work is the effect of open vs. closed eyes on alpha activity. Both conditions are strongly linked in the occipital cortex[58]. Based on our experience, this is not the case in the temporal cortex, which is more related to auditory rather than visual stimuli. However, a formal comparison of both conditions in the auditory cortex is necessary to accurately measure the impact of open vs. closed eyes.

While we focus our work on alpha oscillations, these are not the only rhythms in the auditory cortex[59]. For instance, theta (4–8 Hz) and delta (0–4 Hz) oscillations are key dynamics for speech processing, with the former tracking the syllabic time scale[60] and the latter associated with prosody[61], pauses[62] and other top-down[63] or linguistic processes[64]. Given the nature of our stimuli (short pure tones) and the short intertrial interval ( ~ 1 s), these oscillations are beyond the scope and possibilities of this study. Further work should investigate the interaction between alpha and other brain oscillations, and particularly explore whether, in continuous stimulation contexts such as speech or music perception, the gating of information depends on contextual or predictive features.

Another general limitation inherent to every intracerebral study is the diagnosis of epilepsy. Although it cannot be fully addressed, several precautions can be taken to mitigate its impact. Every trace of epileptic activity was excluded from the analysis, and a partial or complete withdrawal of antiepileptic drug is done prior to the beginning of SEEG exploration, and none of the patients had their epileptogenic zone including the auditory areas. However, functional changes have been detected even in regions non-

involved in the epileptic network[65]. While they mainly affect the broadband connectivity between regions (not studied in this work), we cannot completely rule out any effect in alpha oscillations.

Overall, based on our results we can hypothesize that primary auditory areas have a generally disinhibited state without resting oscillations, ready to process any new stimulus, while there is an inhibitory alpha oscillation who gates or not the information to high-level structures. Both evoked and oscillatory sources have a local bottom-up inhibitory system that becomes active after receiving a new input (induced alpha). Oscillatory sources would have a general inhibitory system that would be steadily suppressed when expecting or processing new information. This sustained suppression also includes the preparatory phase of the task[14], indicating a top-down anticipatory disinhibition prior the stimulus. Finally, the weak high-gamma response and lack of evoked potentials in the oscillatory sources suggests that the sustained alpha rhythm is not restricted to a local inhibition[13] but belongs to a larger network, encompassing primary and non-primary auditory areas.

## Methods

### Participants

A total of 18 patients (10 females) with pharmacoresistant epilepsy were recorded with SEEG during their period of presurgical evaluation at the Hôpital de laTimone (Marseille, France). Neuropsychological assessment carried out before SEEG recordings indicated that all patients had intact language functions and met the criteria for normal hearing. None of them had their epileptogenic zone within the auditory areas as identified by experienced epileptologists. The study was approved in accordance with the Declaration of Helsinki by the Institutional Review board of the French Institute of Health (IRB00003888). Patients provided written informed consent prior to the experimental session. Participation was voluntary, and none of these patients participated in a clinical trial.

### SEEG recordings

SEEG recordings were performed using depth electrodes, implanted stereotactically (Talairach et al.[66]; Alcis, Besançon, France, and Dixi Medical, Chaudefontaine, France). All the patients presented, at least, one electrode in the auditory cortex, implanted orthogonally to the cortical surface, recording the tip of Heschl's gyrus, the planum temporale (Fig. 1a). From the 18 patients, 5 had bilateral implantations, resulting in a total of 23 analyzed electrodes ($N = 23$). The electrodes had between 8 and 18 contacts per electrode, a diameter of 0.8 mm, 2 mm contact length and separated from each other by 1.5 mm. A scalp electrode placed at Fz was used as reference for the recordings. SEEG signal was recorded on a digital system (Natus Medical Incorporated) with sampling at 1024 Hz with 16-bit resolution, a hardware high-pass filter (cutoff = 0.16 Hz), and an antialiasing low-pass filter (cutoff = 340 Hz). To determine the exact location of each electrode and contact, a co-registration of the postimplantation CT-scan with the preoperative MRI was performed for each patient using GARDEL[67] in-house software (https://meg.univ-amu.fr/wiki/GARDEL:presentation). To compare the contact location across subjects in a common space, the patient specific anatomy was warped using the MNI template[68]. From the MNI coordinates, we determined in which region was located each SEEG contact using the Brainnetome atlas[69].

### Implantation and impact of medication

There is no "standard" approach for the electrode implantation, as it is entirely guided by the hypotheses regarding the anatomical location of the epileptogenic zone (EZ). The goal is to identify the specific area for subsequent cortectomy. These hypotheses about the potential location of the EZ are formed based on non-invasive pre-surgical assessments (Phase I), which include a detailed clinical history, surface video-electroencephalographic (EEG) recordings, MRI, and 18FDG-PET scans. Consequently, electrode placement is tailored to each patient's unique clinical profile rather than being standardized.

One of the most commonly explored areas is the perisylvian region, particularly when there is a need to determine whether the patient's epilepsy

is temporal, temporo-perisylvian, or purely perisylvian. Misdiagnosing perisylvian epilepsy is a leading cause of surgical failure in temporal epilepsy. Perisylvian epilepsy can be located in various regions, including the insular cortex, frontal operculum, parietal operculum, temporal operculum, and the superior temporal and supramarginal gyri. In this region, electrodes are typically implanted orthogonally to the cortical surface to capture recordings along the electrode from areas like the posterior insula, the tip of Heschl's gyrus, and the planum temporale. Another approach involves implanting electrodes more anteriorly to target the superior temporal gyrus and ventral insula. The perisylvian region is also explored under the hypothesis of epilepsy originating from the inferior parietal lobule, pericentral area, or ventral prefrontal/premotor cortex.

The transverse gyrus (Heschl's gyrus), which includes the auditory cortex, plays a crucial role due to its connections with lower central regions and the inferior frontal gyrus. This region serves as a pathway through which seizures from the temporal pole and the anterior superior temporal gyrus can spread.

Neural recordings were conducted between 4 and 9 days following the implantation procedure, without the use of sedation or analgesic drugs. Typically, antiepileptic drugs are partially or completely withdrawn before the exploration. However, medication levels are adjusted individually based on the type of seizures. Recordings are consistently taken at least 4 hours after the last seizure.

### Experimental paradigm

We recorded SEEG activity during rest and a pure-tone stimulation task. The resting condition consisted in three intervals of three minutes each, with the patients sitting awake with eyes closed. The auditory stimuli were composed of 100 pure tone trials, 30 ms long, presented binaurally at 1 kHz. The interval between stimuli was 1.030 ( ± 200) ms. We selected this type of stimulus due to its efficiency to activate the auditory cortex without containing linguistic information[16]. Patients were in a sound-attenuated room while passively listening to the pure tones from loudspeakers. Stimuli were presented using E-prime 1.1 (Psychology Software Tools Inc., Pittsburgh, PA, USA). To facilitate the comparison between conditions, we selected only the first 100 s without artifacts during rest, matching with the duration of the auditory task.

### Independent component analysis

ICA aims to solve the 'cocktail party' problem by separating N statistically independent sources that have been mixed on M recording contacts. While the 'cocktail party' problem is often attributed to the identification of one specific acoustic source among many others, in our case we aim to disentangle the different neural sources generating field potentials (i.e., it is not related to the auditory stimuli). It assumes immobility of the neural sources in space, i.e., that the contribution of one source to the recording contacts is the same throughout the recording session. Each recorded signal $u_m(t)$ is thus modeled as the sum of $N$ independent sources ($s_n(t)$) multiplied by a constant factor ($V_{mn}$):

$$u_m(t) = \sum_{n=1}^{N} V_{mn} s_n(t), m = 1, 2, \ldots, M \qquad (1)$$

where $u_m(t)$ is the SEEG data, $V_{mn}$ the ICA weights or spatial profile of each source, $M$ the number of contacts, $N$ the number of sources and $s_n(t)$ the obtained independent components ("SEEG-ICs").

We concatenated the time courses of the selected resting condition window and the whole pure tone recording and ran ICA on each electrode (total: 23 electrodes). Thus, each source had a unique spatial profile, allowing a traceability of the activity of the same neural source across conditions. We obtained as many components as available contacts per electrode (N = M), without any prior dimension reduction[70]. We used FieldTrip[71] to compute ICA based on the infomax algorithm, which aims to minimize the mutual information between components[72], as implemented in EEGLAB[73].

Although not all the SEEG-ICs represented neuronal sources, we did not discard any of them at this point. All the SEEG-ICs were z-scored.

## Analysis of auditory evoked potentials (AEPs)

For each SEEG-IC, we checked whether they had a significant AEP during pure tone stimulation. To do so, we tested if each time point across trials was significantly different from zero with a t-test, obtaining a $t$- and $p$ values for each time-point. Then, we corrected these tests for multiple comparisons using a local false discovery rate (LFDR; Benjamini and Heller, 2007) on the t-values with a threshold of 0.2 (Pizzo et al., 2019). LFDR assumes that the distribution of t-values is Gaussian, considering as significant those values that stand out from the distribution. To have a better estimation of the distribution, we grouped all the t-values across SEEG-ICs of the same electrode, obtaining a single threshold per electrode. To remove artefactual single points, i.e., single data points that were significant but the anterior and posterior samples were not, we selected only those points during the first second after the stimulus and we imposed a minimum number of consecutive significant time samples (10 ms in this work). No baseline correction was applied during the AEP analysis, as the data was already high-pass filtered (0.16 Hz) and z-scored to remove very low frequency trends.

For each SEEG-IC with a significant response, we selected its activation time as the earliest peak in the AEP (see Fig. 4a). This initial peak is generally followed by a much higher response in amplitude. The later response tends to drive the result of most automatic methods, although early activations can be visually detected. Therefore, we used a semi-automatic approach, identifying the local peaks of the AEP using the MATLAB (Mathworks, Natick, MA) function *findpeaks.m*, imposing a minimum prominence of twice the standard deviation of the AEP during baseline (between −100 ms and 0 ms) and considering both positive and negative peaks. We considered as activation time the first peak identified.

## Spectral analysis

Power spectra were estimated using the multitaper method on each SEEG-IC[74] with a frequency resolution of 0.25 Hz. Then, we followed the *fooof* approach to characterize the power of each source[33]. This methodology separates the periodic and the aperiodic (1/f-like) component of the spectra, allowing the analysis of the oscillatory power independently of the changes in the 1/f distribution[75]. The aperiodic component is modeled by a Lorentzian function, where the main parameters are the offset and the exponent (or curvature). Then, the *fooof* method detects each oscillatory peak above the aperiodic components and fits them individually with a Gaussian function, obtaining the power, center frequency and bandwidth of each detected oscillation. We selected the range from 2 to 30 Hz for the *fooof* fit, a minimum bandwidth for peak detection of 0.5 Hz (twice the frequency resolution) and a minimum amplitude of twice the standard deviation of the aperiodic-removed power spectrum (see Supplementary Fig. 1 for a distribution of all the identified peaks across SEEG-ICs and conditions). The knee parameter was fixed at zero.

To limit the contribution of the AEPs during pure tone stimulation to the power spectrum, we also analyzed the time-courses after removing the averaged response from each trial (Pure Tone no AEP condition). First, we computed the AEP for each SEEG-IC. This averaged AEP was then fitted for each trial using a linear regression minimizing the difference between the single trial and the fitted AEP and it was subtracted for each trial. In other words, we subtracted to each trial the averaged AEP multiplied by a factor 'k' that minimized the result. The remaining time-course should contain all the activity that is not explained by the evoked response. The spectral analysis was computed on the resultant time-course.

## Identification of sources (independent components of interest)

We visually inspected all the SEEG-ICs to remove the components related to the reference or to remote sources, i.e., with a similar activity along all the contacts of the electrode (Fig. 1d, gray component). Then, we classified the remaining components as "oscillatory" sources if they presented a high alpha oscillatory activity during rest (i.e., a significant peak of power given by

the *fooof* analysis), "evoked" sources if they had a significant AEP, both "oscillatory" and "evoked" sources, or "nonrelated" components. A total of 58 SEEG-ICs were labeled as evoked. To determine the oscillatory SEEG-ICs, we selected, for each source, the highest significant peak of the periodic component at 5−10 Hz obtained from the power spectrum during rest. From all the SEEG-ICs, we chose the 25% with highest power (71 SEEG-ICs; 0.75 quantile), labeling them as oscillatory. As one of the goals of this study was to test whether the sources processing the input (i.e., evoked sources) were also the sources with the main alpha oscillatory activity in rest, we fixed this value to include SEEG-ICs from most of the electrodes in the analysis (21 out of 23 electrodes had an oscillatory source) while keeping a similar number of oscillatory and evoked source (71 versus 58, respectively). While the classification of a source as "evoked" is relatively independent of other SEEG-ICs, we determined the "oscillatory" sources given the total number of SEEG-ICs in the analysis. Therefore, we repeated our analysis using quantiles 0.9 and 0.5 to ensure that our selection criterion was not driving the results.

To check that none of the conditions were biasing the resulting SEEG-ICs, we repeated the ICA on each task separately and compared the obtained components with the original sources. First, we performed an ICA and extracted a new set of components for each condition (condition-specific SEEG-ICs). Second, we split the time-courses of the original SEEG-ICs in the two conditions to match the duration of the condition-specific SEEG-ICs. Then, we computed the correlation between the original SEEG-ICs and the condition-specific SEEG-ICs, selecting the maximal correlation, i.e., the most similar component, as it should reflect the same source. Therefore, for each oscillatory and evoked source we obtained two correlation values. One related to the most similar condition-specific SEEG-IC during rest, and another related to pure tone stimulation. If both correlation values are similar and close to 1, it suggests that the same source is active during both conditions. On the contrary, is the values are maximal in one condition but close to zero in the other, this would indicate that the source is active only during that condition, which would guide the joint analysis.

## Time-frequency analysis

In addition to the AEP, we tested whether each SEEG-IC had a response to the pure tone stimulation in the time-frequency plane. We used Morlet wavelets to obtain the time-frequency transformation with 7 cycles per wavelet. Then, the time-frequency responses were averaged across trials for each SEEG-IC and they were baseline corrected using the z-score. The baseline was set between 200 and 300 ms before the stimulus to avoid temporal smearing from post-stimulus activity.

To compute the statistical differences with baseline at the group level, either across oscillatory SEEG-ICs or evoked SEEG-ICs, we performed a surrogate analysis followed by a cluster-based correction[76,77]. First, we selected the averaged response of each SEEG-IC and we computed a t-test across sources between each time-frequency point after the stimuli against the baseline at the same frequency. We defined a cluster of significance as a group of adjacent time-frequency points with a significant $p$-value (non-corrected $p$-value lower than 0.01 in this work). We assigned to each cluster the sum of the t-values within the cluster (either positive or negative). For each surrogate ($N = 1000$), we randomly shifted the time of the stimuli in a window of ±500 ms around the original value. This way, the time-frequency response remained the same, but the temporal alignment was broken. We repeated the cluster procedure for each surrogate, keeping both the clusters with maximal and minimal t-value at each iteration. Any significance found in these surrogates would be by chance. Finally, we tested whether the t-values of our original clusters were significantly higher than the maximal t-values of the surrogate analysis for positive effects or lower than the minimum for negative effects. The threshold of significance was set at the 97.5 percentile of the distribution of surrogate values ($p$-value < 0.025), including the observed value into the simulated values to avoid $p$-values equal to zero[78].

The statistical comparison of the time-frequency responses of the oscillatory SEEG-ICs versus the evoked the SEEG-ICs was performed using

a t-test. We computed the statistical test across sources between each time-frequency point of the oscillatory SEEG-ICs and the same point of the evoked SEEG-ICs. Then, we corrected all the *p*-values using FDR.

As a single patient may have several SEEG-ICs and, therefore, drive the results, we repeated the time-frequency analysis averaging, for each subject, all the oscillatory or evoked sources. This way, only one averaged oscillatory source and one evoked source were considered in the analysis, obtaining a number of observations equal to the number of participants with, at least, one significant oscillatory or evoked SEEG-IC.

### Inter-trial phase clustering

We analyzed how much of the time-frequency activity was related to the presence of the AEP using the inter-trial phase clustering (ITPC; Cohen, 2014). For each time point and frequency, it measures the distribution of phases across trials as:

$$ITPC(t,f) = \left| \frac{1}{N} \sum_{n=1}^{N} e^{i\varphi_{ntf}} \right| \qquad (2)$$

where $\varphi_{ntf}$ is the phase of the signal at trial $n$, time $t$ and frequency $f$. If the distribution is centered around a specific phase, i.e., it is consistent across trials, then the ITPC value would be close to one. On the hand, a uniform distribution would yield an ITPC value close to zero.

We also tested whether the instantaneous phase at stimulus arrival conditioned the brain response[79]: either the amplitude of the AEP or the changes of power at high gamma (80-120 Hz) and alpha (5-10 Hz) frequencies. To do so, we used a weighted ITPC (wITPC):

$$wITPC = \left| \frac{1}{N} \sum_{n=1}^{N} b_n e^{i\varphi_{\alpha n}} \right| \qquad (3)$$

where $\varphi_{\alpha n}$ was the alpha (6-10 Hz) phase at stimulus arrival for the trial $n$, and $b_n$ were the different factors to test (amplitude of the AEP, gamma and alpha power).

The amplitude of the AEP for each trial was obtained from the linear model used to remove the contribution of the evoked response (factor 'k', see *Spectral analysis* section). The gamma power was obtained as the maximal value in the time-frequency map of each trial, using a time from 0 to 200 ms, and a frequency bandwidth from 80 to 120 Hz. The same approach was followed for the alpha power, but selecting the minimum value (i.e., the highest alpha suppression) in the time window from 100 to 300 ms and the frequency band from 0 to 20 Hz.

The statistical analysis of the wITPC was done using permutation testing[76]. On each iteration ($n = 1000$) the $b_n$ values were randomly distributed across trials breaking the relationship between phases and factors and creating a null hypothesis distribution. Then the z-scored of the measured wITPC is computed by subtracting the mean of the null hypothesis and then dividing by the standard deviation. Finally, z-scored wITPC values higher than 1.96 (95% confidence interval) were considered significant.

### Onset latency of gamma activity

We measured the event-related gamma power by averaging, for each trial, the baseline corrected time-frequency map between 80 and 120 Hz, obtaining a single time course for each SEEG-IC. Then, we computed whether this activity was significantly different from zero following the same approach as for the auditory evoked potentials. Finally, the onset latency of each source was considered as the first time point of the event-related power significantly different from zero and with a minimal duration of 30 ms[44].

### Statistical analysis

To compare whether the location of the oscillatory and evoked SEEG-ICs is different, we selected for each source its location on the x-axis (lateral-medial axis) using the MNI coordinates. This dimension was selected as it

corresponds to the trajectory of the implanted electrodes, allowing a good sampling from medial to lateral locations and intra-electrode comparisons. First, we compared with a t-test whether the absolute values of the coordinates were different between sources. Then, we measured whether the location of the sources was distributed or local by computing the variance of the location in the lateral-medial axis. A large variance would indicate that the sources are distributed along the axis, while a small value would suggest that the sources are located in a specific location. We tested whether the standard deviation of the location in the oscillatory SEEEG-ICs was different from the evoked SEEG-ICs using a permutation test. For each of $n = 1000$ permutations, we randomly labeled each SEEG-IC as oscillatory or evoked, preserving the original ratio. We then computed the difference between the standard deviation of the location of the permuted oscillatory SEEG-IC and the permuted evoked SEEG-IC. Finally, we counted on how many permutations the obtained difference was higher than the original value. This value was divided by the total number of permutations plus the observed value ($n = 1001$) to obtain the *p*-value[76,78].

To test how the dynamics of the same SEEG-IC vary across conditions (rest versus pure tone), we tested different features of the power spectrum (peak frequency, relative power and aperiodic component) using a paired t-test. To test whether the different features vary across the lateral-medial axis of the auditory cortex, we computed the Pearson correlation between the x coordinates of each sources and the value of the different features.

### Reporting summary

Further information on research design is available in the Nature Portfolio Reporting Summary linked to this article.

### Data availability

The conditions of our ethics approval do not permit public archiving of anonymized study data. Readers seeking access to the data should contact the lead author. Access will be granted to named individuals in accordance with ethical procedures governing the reuse of clinical data, including completion of a formal data sharing agreement.

### Code availability

All the code used in this work is available at https://github.com/VictorLopezMadrona/Matlab_analysis_neuroscience. The numerical source data for all the figures is available at[80].

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

## Acknowledgements

We would like to thank Manuel Mercier for assisting with the data collection and the development of the computational framework for localizing the SEEG electrodes in common patient space, Paul Lalande-Robert for his help on the visualization of the results and Aurélie Bidet-Caulet for helping with the interpretation of the results. This study was supported by a postdoctoral fellowship from the Institute of Language, Cognition and the Brain (ILCB, grant ANR-16-CONV-0002). This work has received support from the ANR-20-CE28-0007-01 (to B.M), ANR-21-CE28-0010 (to D.S.), ANR-17-EURE-0029 (NeuroMarseille), Fondation Pour l'Audition (FPA RD-2022-09), and co-funded by the European Union (ERC, SPEEDY, ERC-CoG-101043344). It was also supported by the French government under the Programme «Investissements d'Avenir», and the Excellence Initiative of Aix-Marseille University (A*MIDEX, AMX-19-IET-004).

## Author contributions

All authors Designed Research; V.L.M., A.T., B.M. Performed Research; V.L.M., C.G.B. Analyzed Data; V.L.M., D.S., B.M. Interpreted results; V.L.M. Wrote the first draft of the paper; all authors contributed to the manuscript final writing.

## Competing interests

The authors declare no competing interests
