## [Transparent Peer Review file · Communications Biology]

Different sustained and induced alpha oscillations emerge in the human auditory cortex during sound processing

Corresponding Author: Dr Victor Lopez-Madrona

Version 0:

Reviewer comments:

Reviewer #1

(Remarks to the Author)

Madrona et al. use intracerebral recordings from the auditory cortex of epilepsy patients to describe two separate functional mechanisms when listening to pure auditory tones. Using ICA in concatenated data from resting and listening conditions, they find spatially segregated sources along the medial-lateral axis: a) 'oscillatory' alpha components and b) 'evoked' components. The 'oscillatory' ones are traced predominantly laterally, are active during rest and strongly attenuated during sound processing. The 'evoked' components are located mostly medially, are silent during rest and show a clear evoked response with a sustained alpha suppression. The manuscript is straightforward, well motivated and for the most part clearly written (I specifically compliment the authors for the comprehensive methods section) while it provides important information regarding alpha inhibition mechanisms during audition. Few points for the otherwise sound and solid paper:

Frequency resolution in the spectral analysis can affect the foof fitting, and the frequency peaks identification. What was the resolution of the spectral analysis (line 180)?

Line 189: why remove the evoked response from the single trials, when the focus is on alpha (and higher frequencies)? Are the results presented reproducible without the removal of the AEPs?

The time-frequency plots (fig 3b, 4f and 5) exclude the lower-frequencies. While the TF plots are similar for the oscillatory and evoked components, I would expect some differences in the <10 Hz range, which would be interesting to show. It is also not clear if for this analysis the AEPS were removed from the trials or not.

Line 490 needs citations

Line 177: what were the parameters of the findpeaks function?

Reviewer #2

(Remarks to the Author)

The paper presents the analysis of intracranial EEG recordings with depth electrodes in the human auditory cortex, and compares conditions of rest and auditory stimulation. The signal processing pipeline separates components obtained through ICA into evoked and oscillatory features.

Statistical analyses of the components show that there is predominantly evoked activity during stimulation, and more alpha activity during resting state. Further analyses of the evoked response in the time-frequency domain reveal induced gamma band responses a reduction of alpha power following the presentation of tones.

The clear strength of the work lies in the presentation of a rare and valuable dataset, as invasive recordings are not commonly available in humans. Novel and detailed insights can be gained from the approach, and in principle the analyses are suited to do so. However, I am currently not fully convinced that the analyses fully support the conclusions, notably because some of the choices made throughout are not well justified in light of the hypotheses, and some methodological details are missing.

1) My main point is that strong support statements as found in the discussion, are currently not well aligned with the results, for instance:

I. 444 "In this work, we have defined two distinct sources of activity in the auditory cortex: one presenting high alpha-like oscillations (5-10 Hz) during resting state that are suppressed during auditory stimulation (oscillatory source); and a second one that is relatively silent at rest but presents a significant evoked response during pure-tone stimulation (evoked source)."

I. 482: "During the presentation of pure tones this rhythm is strongly suppressed"

I. 538: "Overall, based on our results we can hypothesize that primary auditory areas have a generally disinhibited state without resting oscillations, ready to process any new stimulus, while 539 there is an inhibitory alpha oscillation who gates or not the information to high-level structures."

These statements appear a bit of a stretch, as there is no clear link to the results provided. In particular terms like 'suppression' or 'disinhibition' would require an analysis that shows that establish first an independent assessment of the sources of activity, and then link them across conditions. The way the ICA components were computed and selected (data from both conditions combined, selection to obtain equal numbers of evoked and oscillatory components) bears the risk of biasing the counts and hence the conclusions that rely on statistics performed on the components and counts. It is possible that the most prominent components are still driven by the auditory stimulation, as spontaneous activity might be more variable, and therefore more of the resulting components come from the stimulation period.

A complementary approach would be to compute ICA components separately for both conditions and compare the oscillatory ones between conditions.

To test whether certain sources are active in both conditions, components could then be reciprocally projected onto the other condition. It seems that such projections have been computed for the data displayed in Figure 1 b/e, 3b/c, 4d/e, where single or multiple components were labelled as 'resting' or 'evoked', but the analysis does not allow to independently assess which condition the component came from in the first place.

It would also be helpful to report more descriptive information about the ICs, for instance the spectra and distributions of peaks per condition. Were there components with more than one marked peak, or peaks anywhere else than in the alpha range?

2) I was lacking a clear hypothesis in the introduction to justify the analysis choices, and to guide the reader through the results.

It should be clearly stated what is expected in comparing the two conditions, and whether the a-priori rationale was to focus on specific frequency bands (alpha) in the oscillatory analysis (confirmatory) or rather exploratory.

3) What about lower frequent oscillations given the relatively rhythmic auditory stimulation? The spectral analysis does not seem well suited to extract peaks in lower frequency regions due to the 1/f property of the spectrum (in FOOOF the lower frequency cutoff strongly determines whether low frequent peaks are observed or not), and also the removal of the average response might have removed phase-locked oscillations (also see point on evoked / induced below).

4) The use of ICA to select components of interest is put forward as an important strength of the work (l. 80-84), or even a goal, but I was missing a clear link to the scientific goals / hypotheses.

It is per se not so novel to use ICA for selection of components of interest rather than rejection of artifacts (see e.g. Chaumon & Busch 2014).

Please clarify why the ICA has particular benefits for sEEG and why it can "remove the electrical reference and retrieve remote sources not sampled" (l. 80-84)?

5) I. 492: "Our work complements this result by describing two types of alpha suppression (sustained and induced). The evoked sources, mainly related to primary posteromedial structures, presented low alpha power during rest." No clear hypotheses were outlined concerning evoked or induced responses. Removing the average evoked response before computing time-frequency analyses also removes time-locked oscillations, which might occur due to phase resets in the stimulation condition, both in the alpha band and in lower frequency bands (delta).

5) If I understood correctly, participants had their eyes closed during the resting state, but not during the auditory stimulation. Previous work has shown that there might be different sources of alpha underlying these two states (Wöstmann et al. 2020), could this have an effect on divergence of alpha sources between conditions?

6) Could you address in the discussion whether the fact that all participants are diagnosed with epilepsy might have any influences on the findings?

7) The methods lack some important details:

- AEP analyses: was a baseline applied before testing against 0?

Were the SEEG-ICs also z-scored for the analysis of evoked responses or just for computing power spectra?

- l.190/191: 'This averaged AEP was then fitted for each trial using a linear regression minimizing the difference between the single trial and the fitted AEP. Finally, we subtracted the fitted AEP 191 independently of each trial.'

I did not understand this part and how it produces a single trial signal to subtract from single trials. Please clarify.

- I 197: "We visually inspected all the SEEG-ICs to remove the components related to the reference or to remote sources, i.e., with a similar activity along all the contacts of the electrode (Figure 1d, 198 gray component). Then, we classified the remaining components as "oscillatory" sources if they presented a high oscillatory activity during resting state, "evoked" sources if they had a significant 200 AEP, both "oscillatory" and "evoked" sources, or "nonrelated" components." What does it mean for a component to present 'high oscillatory activity', and were there any measures taken to prevent biases by the raters?

- Please provide the parameters used in the FOOOF analysis.

- In the following the authors state that the highest significant peak was selected and then the 25% ICs with highest power. The rationale for choosing 25% sounds a bit handwavy, and might bias the results, as there could be a priori more or less oscillatory compared to evoked components.

Even though a control analysis was performed with different percentages, it seems to be a strong assumption to a-priori limit the number of oscillatory components. This might obscure oscillations in different frequency bands. As indicated above, it was not clear to me whether the approach was to solely focus on alpha oscillatory components, in which case the analyses could be more tailored towards this frequency band and its variations.

- I.254/255: It sounds important to first test whether the location is different, before addressing the standard deviation? The standard deviation could be biased by the location of the contact, for instance if it is close to the edge of the electrode.

References:

Chaumon M, Busch NA. Prestimulus neural oscillations inhibit visual perception via modulation of response gain. *J Cogn Neurosci*. 2014 Nov;26(11):2514-29. doi: 10.1162/jocn_a_00653

Wöstmann, M., Schmitt, L. M., & Obleser, J. (2020). Does closing the eyes enhance auditory attention? Eye closure increases attentional alpha-power modulation but not listening performance. *Journal of cognitive neuroscience*, 32(2), 212-225.

Reviewer #3

(Remarks to the Author)

In this paper, the authors use intracranial EEG and independent component analysis to describe two dissociable types of activity in the superior temporal lobe, one that oscillates at rest and does not display low-frequency evoked responses to sounds, and another with the opposite characteristics. The methods are appropriate and well-described, the results are described and illustrated eloquently, and the discussion is interesting and very reasonable. Overall, this paper represents a focused, important contribution to our understanding of how the brain's activity organizes at rest and changes during the processing of sensory inputs. I only have a few suggestions for the authors to entertain.

Major comments:

Re: the existence of two kinds of alpha suppression (induced and sustained): I would like to suggest that the authors explore the phase consistency of both types of alpha activity with respect to the auditory stimulus. An absence of inter-trial phase consistency of sustained alpha drop would be an additional suggestion that something non-specific is at play. Similarly, I would be curious to know about the impact of instantaneous phase at stimulus arrival on all types of stimulus-locked responses (low-frequency evoked potentials and broadband high-frequency power changes). The phase of ongoing oscillations in auditory cortex likely modulates its responsiveness to incoming stimuli (see e.g. Schroeder et al., *TICS* 2008; Thézé et al., *Sci Adv* 2020; and Ahveninen et al., *J Neurosci* 2024, in the multisensory context).

Re: high-frequency power changes at both oscillatory vs. evoked sites: power is clearly lower in oscillatory sites, but what about onset latency? Finding a latency difference between the two types of response might help establish a processing hierarchy.

Minor points:

General note: would the authors consider replacing "state" (as in resting state) by "task" or "condition", in order to separate their two conditions? To me, "state" evokes varying states of vigilance (wakefulness vs. drowsiness, sleep, anesthesia, coma) rather than the particulars of what an awake and behaving person is asked to do (even when they are asked to not do anything in particular).

Introduction:

A quick comment re: resting iEEG activities: in addition to the Frauscher et al., 2018 reference, you might consider consulting Groppe et al., *Neuroimage* 2013, who basically did the same thing earlier.

Methods:

Description of the selected SEEG electrodes (Methods > SEEG recordings and Figure 1 a): was the selection of patients and electrode shafts driven by the presence of one or more contacts in a given parcellation of the Brainnetome atlas, or else done by hand (visual inspection)? What were the regions targeted by the electrodes that passed through Heschl's gyrus or the planum temporale? From Figure 1a, I have the impression that the deep-most contact is in the thalamus, which is not a typical target of invasive EEG recordings for epilepsy surgery planning.

Listening task (Methods > Experimental paradigm): was the listening done with eyes closed, like the resting state? If not, could this introduce bias?

Regarding the x-axis (Methods > Statistical analysis): would it be easier for the reader if the text referred to the lateral-medial axis?

Regarding the permutation test on the variance in the x-axis location of oscillatory vs. evoked ICs (Methods > Statistical analysis): a small thing really, but when using permutation testing to compute the probability that a variable be observed by chance under the null hypothesis, the observed value of the variable must be added to the set of simulated values. The computation of the p-value is then the sum of the N simulated values and the observed one that are larger or equal to the observed value, divided by the number of simulated values + 1. That way, the smallest possible p-value is not 0/N, which is obviously problematic, but $1/(N+1)$. See e.g. the book by Manly (1997), or Phipson and Smyth, *Stat Appl Genet Mol Biol* 2010 (<https://doi.org/10.2202/1544-6115.1585>).

Results:

Example recordings (Figure 1b,c): given the high SNR of iEEG, would it be possible to show a single-trial AEP on panel c, rather than an average? It would allow the reader to grasp how oscillatory activity (depicted on panel b) attenuates during stimulus processing.

Location of ICs (Figure 2b): I suggest adding orientation labels to the views of the temporal lobes, indicating where the pole, planum, medial and lateral sides are.

Discussion:

I would like to suggest that the authors briefly explore how to test the hypothesis that they lay out in the last paragraph of the discussion (earlier auditory cortex being always "on" vs. the activity of higher-order areas being gated by resting oscillations). With continuous stimuli? or a more complex sequence of auditory stimuli that are more or less predictable of what comes next?

Pierre Mégevand

Version 1:

Reviewer comments:

Reviewer #1

(Remarks to the Author)

I would like to congratulate the authors for the extensive effort on revising the manuscript. They have have satisfactorily addressed my comments and concerns.

Reviewer #2

(Remarks to the Author)

The authors provided an extensive revision of their paper, including novel analyses, a newly recorded dataset (single participant, not included in the manuscript), and thorough revisions. Overall, these make the paper much clearer and I believe that the work will be very appreciated in the field. I have no further comments.

Reviewer #3

(Remarks to the Author)

I thank the authors for their rebuttal and revision. They have addressed my comments adequately. I have 2 last questions for them:

No changes in power spectrum in temporal cortex when eyes are open and when they are closed: given that another reviewer also raised this question, and even though I understand that the authors will not want to add a figure with a single patient's data to their manuscript, they should still consider adding a sentence to the paper, stating that (for instance) based

on their experience, no power change is expected in temporal cortex when eyes are open vs. close.

Targeting of S-EEG electrodes: I was not questioning the fact that superior temporal (incl. auditory) cortex is implanted en route to the insula, but rather the fact that the S-EEG electrode reaches all the way to the thalamus. As the authors say themselves, the goal of a S-EEG implantation is to plan a tailored cortical resection. The thalamus is never a target of resective epilepsy surgery. Additionally, the implantation of the thalamus is not mentioned in the French S-EEG guidelines (<https://www.sciencedirect.com/science/article/pii/S0987705317302873>). So, the implantation of a S-EEG electrode to the thalamus is not standard clinical practice, at least in most centers. This is what I think needs to be justified.

Pierre Mégevand

Reviewer #1 (Remarks to the Author):

Madrona et al. use intracerebral recordings from the auditory cortex of epilepsy patients to describe two separate functional mechanisms when listening to pure auditory tones. Using ICA in concatenated data from resting and listening conditions, they find spatially segregated sources along the medial-lateral axis: a) 'oscillatory' alpha components and b) 'evoked' components. The 'oscillatory' ones are traced predominantly laterally, are active during rest and strongly attenuated during sound processing. The 'evoked' components are located mostly medially, are silent during rest and show a clear evoked response with a sustained alpha suppression. The manuscript is straightforward, well motivated and for the most part clearly written (I specifically compliment the authors for the comprehensive methods section) while it provides important information regarding alpha inhibition mechanisms during audition. Few points for the otherwise sound and solid paper:

We thank the reviewer for the positive assessment of our work.

Frequency resolution in the spectral analysis can affect the foof fitting, and the frequency peaks identification. What was the resolution of the spectral analysis (line 180)?

The resolution was 0.25 Hz. This information has now been added in the Methods section (p. 10):

"Power spectra were estimated using the multitaper method on each SEEG-IC (Thomson, 1982) with a frequency resolution of 0.25 Hz."

We also have rerun this analysis with an increased resolution (0.05 Hz), to investigate whether the frequency of the peaks changed between conditions, and obtained an equivalent result (no differences in frequencies between resting and pure tone conditions).

We have included this new analysis in the Result section, p.20:

"Since frequency resolution can impact the quality of the fit in the FOOF algorithm, we repeated the analysis with a higher resolution (0.05 Hz; original resolution = 0.25 Hz; see Methods) with equivalent results (frequency peak: 7.9 ± 1.06 Hz and 7.9 ± 1.09 Hz, mean \pm s.d. for rest and pure tone conditions; paired t-test, $p > 0.7$, $t = -0.38$, $df = 59$, $CI = -0.08, 0.06$, effect size = -0.01)."

Line 189: why remove the evoked response from the single trials, when the focus is on alpha (and higher frequencies)? Are the results presented reproducible without the removal of the AEPs?

We removed the AEP because its waveform is mainly composed by ~10 Hz activity, driving the time-frequency map for low frequencies (see e.g. Morillon et al., 2012 <https://doi.org/10.3389/fpsyg.2012.00248>, for a similar description). We have included a new figure (new Figure 5) comparing the time-frequency with and without AEP, as well as the effect in the ITPC. While most of the activity at low frequencies and between 0 and 200 ms is explained by the AEP, the effect in alpha suppression and high gamma activity remains. Results p. 25:

“The time-frequency response of the evoked SEEG-ICs was also driven by the AEP (Figure 5). The waveform of the auditory response had a ~10 Hz pattern and most of the activity at low frequencies was strongly phase-locked with the stimulus (Figure 5a, right panel). The AEP also created a chimney effect in the time-frequency spectrum, close to stimulus onset. Therefore, neural activity in the entire frequency range was affected by the AEP and cannot be directly associated with changes in the oscillatory dynamics. To mitigate this effect, we also analyzed the response after removing the averaged evoked response (AEP; i.e., the phase-locked activity) from each trial (see Methods). Without the AEP contribution, the time-frequency map of the evoked sources had some similarities with the oscillatory sources (Figure 5b vs. Figure 3e). There was an early activation (here at very low frequencies; ~10 Hz) with a strong high-gamma response that started 25 ms after stimulus onset and lasted 200 ms, followed by a suppression at low frequencies (between 5 and 30 Hz), starting at 125 ms in these sources ($p < 0.05$, corrected with FDR). We repeated the analysis by averaging all the evoked SEEG-ICs per subject (number of observations equal to the number of patients with an evoked source), obtaining the same time-frequency pattern (Supplementary figure 3).

Figure 5: Time-frequency and phase locking responses of evoked SEEG-ICs:

- a) Averaged time-frequency response of evoked SEEG-ICs during pure tone stimulation (left panel). Framed areas represent clusters of significant modulation of activity compared to baseline (-300 -200 ms; $p < 0.01$, surrogate test). Inter-trial phase clustering (ITPC) representing the frequencies phase-locked with the stimulus (right panel)
- b) Same analysis as in panel a but after removing the contribution of the averaged AEP to the evoked SEEG-IC time-courses. This process removes only the activity phase-locked with the stimulus.

The time-frequency plots (fig 3b, 4f and 5) exclude the lower-frequencies. While the TF plots are similar for the oscillatory and evoked components, I would expect some differences in the <10 Hz range, which would be interesting to show. It is also not clear if for this analysis the AEPs were removed from the trials or not.

We have increased the range of TF plots up to 2 Hz. Unfortunately, given the short interval between stimuli (~1s), low frequencies (< 4 Hz) were difficult to explore in our task, and the wavelet decomposition may also be affected by the response in consecutive trials. We have added a new paragraph on the limitations of low frequencies in the Discussion section (p. 31):

“While we focus our work on alpha oscillations, these are not the only rhythms in the auditory cortex (Mai et al., 2016). For instance, theta (4-8 Hz) and delta (0-4 Hz) oscillations are key dynamics for speech processing, with the former tracking the syllabic time scale

(Luo and Poeppel, 2007) and the latter associated with prosody (Inbar et al., 2023), pauses (Chalas et al., 2023) and other top-down (Fontolan et al., 2014) or linguistic processes (Chalas et al., 2024). Given the nature of our stimuli (short pure tones) and the short intertrial interval (~1s), these oscillations are beyond the scope and possibilities of this study. Further work should investigate the interaction between alpha and other brain oscillations, and particularly explore whether, in continuous stimulation contexts such as speech or music perception, the gating of information depends on contextual or predictive features.”

Line 490 needs citations

We have added the corresponding citation:

Billig, A.J., Herrmann, B., Rhone, A.E., Gander, P.E., Nourski, K.V., Snoad, B.F., Kovach, C.K., Kawasaki, H., Howard, M.A., Johnsrude, I.S., 2019. A Sound-Sensitive Source of Alpha Oscillations in Human Non-Primary Auditory Cortex. J. Neurosci. 39, 8679–8689. <https://doi.org/10.1523/JNEUROSCI.0696-19.2019>

Line 177: what were the parameters of the findpeaks function?

We have included this information in the Methods section (p.9)

“We identified the local peaks of the AEP using the MATLAB (Mathworks, Natick, MA) function findpeaks.m, imposing a minimum prominence of twice the standard deviation of the AEP during baseline (between -100 ms and 0 ms) and considering both positive and negative peaks. We considered as activation time the first peak identified.”

Reviewer #2 (Remarks to the Author):

The paper presents the analysis of intracranial EEG recordings with depth electrodes in the human auditory cortex, and compares conditions of rest and auditory stimulation. The signal processing pipeline separates components obtained through ICA into evoked and oscillatory features.

Statistical analyses of the components show that there is predominantly evoked activity during stimulation, and more alpha activity during resting state. Further analyses of the evoked response in the time-frequency domain reveal induced gamma band responses a reduction of alpha power following the presentation of tones.

The clear strength of the work lies in the presentation of a rare and valuable dataset, as invasive recordings are not commonly available in humans. Novel and detailed insights can be gained from the approach, and in principle the analyses are suited to do so. However, I am currently not fully convinced that the analyses fully support the conclusions, notably because some of the choices made throughout are not well justified in light of the hypotheses, and some methodological details are missing.

1) My main point is that strong support statements as found in the discussion, are currently not well aligned with the results, for instance:

I. 444 "In this work, we have defined two distinct sources of activity in the auditory cortex: one presenting high alpha-like oscillations (5-10 Hz) during resting state that are suppressed during auditory stimulation (oscillatory source); and a second one that is relatively silent at rest but presents a significant evoked response during pure-tone stimulation (evoked source)."

I. 482: "During the presentation of pure tones this rhythm is strongly suppressed"

I. 538: "Overall, based on our results we can hypothesize that primary auditory areas have a generally disinhibited state without resting oscillations, ready to process any new stimulus, while 539 there is an inhibitory alpha oscillation who gates or not the information to high-level structures."

These statements appear a bit of a stretch, as there is no clear link to the results provided. In particular terms like 'suppression' or 'disinhibition' would require an analysis that shows that establish first an independent assessment of the sources of activity, and then link them across conditions. The way the ICA components were computed and selected (data from both conditions combined, selection to obtain equal numbers of evoked and oscillatory components) bears the risk of biasing the counts and hence the conclusions that rely on statistics performed on the components and counts. It is possible that the most prominent components are still driven by the auditory stimulation, as spontaneous activity might be more variable, and

therefore more of the resulting components come from the stimulation period.

A complementary approach would be to compute ICA components separately for both conditions and compare the oscillatory ones between conditions.

To test whether certain sources are active in both conditions, components could then be reciprocally projected onto the other condition. It seems that such projections have been computed for the data displayed in Figure 1 b/e, 3b/c, 4d/e, where single or multiple components were labelled as 'resting' or 'evoked', but the analysis does not allow to independently assess which condition the component came from in the first place.

The reviewer is correct that by combining the two (rest and pure tone) conditions to perform the ICA analysis, one condition could drive the results. We have thus performed new analysis and included more information to better relate our independent components to the sources of activity on each condition.

First, we have included the total explained variance of each SEEG-IC (i.e., the contribution of each component to the whole recordings). Our results indicate that the oscillatory sources account for a higher explained variance than the evoked sources, suggesting that spontaneous activity was substantial enough for our findings not to be solely driven by the stimulation period. We now state in the Results section, p.18:

“The explained variance of the SEEG-ICs (i.e., contribution of the component to all the SEEG recordings) were different for both sources. Each oscillatory source explained between 0.4% and 63% (mean 10.5%) of the total activity of the data, while evoked sources captured between 0.4% and 27.5% (mean 6.4%) of the variance, suggesting that spontaneous activity was predominant in our recordings.”

A limitation of ICA is that we cannot directly use the same matrix in a different time window (i.e., we cannot use the ICA matrix from rest to identify the sources during pure tone stimulation). This is because the unmixing matrix to disentangled the components is composed by the neural sources and the noise. The neural sources should be fixed in space (i.e., same spatial profile as in Figure 1d), however the noise is different at each time-window (different spatial profiles). In other words, when applying the ICA matrix in a different condition, it will assume that the noise is the same.

However, in order to overcome this limitation, we have performed a complementary approach to compare the ICA between conditions. First, we computed a new ICA on each condition separately. Then, we computed the correlation between the time-course of the oscillatory and evoked components estimated in the combined ICA analysis and the components estimated from each condition separately. If for a given oscillatory/evoked source there is a component in rest and another in pure tone stimulation which is highly correlated with it, then we can conclude that the source is active during both conditions.

Our results show that both types of sources are quite stable across components, although the evoked sources were more difficult to retrieve when analyzing the rest condition alone. We now state in the Results section, p.18:

“To further check that the obtained sources were not driven solely by one of the two conditions, we repeated the ICA procedure on each condition separately and computed the correlation between the previous SEEG-ICs (with both conditions concatenated) and the new time courses (computed separately for both conditions; see methods). The oscillatory components were quite stable across analyses, and the same components were retrieved when ICA was computed only on the rest condition (averaged correlation of SEEG-ICs \pm standard deviation, s.d.: 0.84 ± 0.13) or only during the stimulation condition (0.84 ± 0.14). Results were similar for the evoked sources, although it was more difficult to retrieve them when analyzing only the rest condition (averaged correlation of SEEG-ICs: 0.76 ± 0.13), compared to the pure tone condition (0.81 ± 0.15). Therefore, the two types of sources were present at both rest and during stimulation, rather than being active only in one condition and completely silent during the other.”

It would also be helpful to report more descriptive information about the ICs, for instance the spectra and distributions of peaks per condition. Were there components with more than one marked peak, or peaks anywhere else than in the alpha range?

We have included a new figure (Supplementary Figure 1) with the distribution of peaks of all SEEG-ICs across conditions, including their relative amplitude. There were two clear clusters of peaks, in the alpha range and its first harmonic. Moreover, the alpha components were those exhibiting higher relative power.

Supplementary Figure 1: Distribution of peaks identified with the foof approach for all the SEEG-ICs. There are two clear clusters of peaks, centered around 8 Hz and its first harmonic 16 Hz. The most prominent oscillations (i.e., peaks with higher relative power) were also centered between 5 and 10 Hz.

2) I was lacking a clear hypothesis in the introduction to justify the analysis choices, and to guide the reader through the results.

It should be clearly stated what is expected in comparing the two conditions, and whether the a-priori rationale was to focus on specific frequency bands (alpha) in the oscillatory analysis (confirmatory) or rather exploratory.

We apologize the lack of clarity regarding our scientific goals and choices.

Our main question was to address whether the alpha inhibition was intrinsic to each neural source (i.e., the sources processing the stimulus also present alpha oscillations that modulate the response) or extrinsic (there are neural sources processing the input without alpha activity and different sources with alpha oscillations that control the inhibition of the whole network).

We have rephrased the abstract accordingly:

“Alpha oscillations in the auditory cortex have been associated with attention and the suppression of irrelevant information. However, their anatomical organization and interaction with other neural processes remain unclear. Do alpha oscillations function as a local mechanism within most neural sources to regulate their internal excitation/inhibition balance, or do they belong to separated inhibitory sources gating information across the auditory network? To address this question [...]”

And further clarified our goal in the Introduction (p.4 and 5):

“During speech processing, alpha activity is suppressed in regions responding to the stimulus, suggesting a local bottom-up disinhibition to favor information processing (Müller and Weisz, 2012; Strauß et al., 2014). On the other hand, alpha activity has also been associated to top-down anticipatory processes (Müller and Weisz, 2012), which may require sources located in higher level areas and to control the excitation/inhibition balance in other regions. In this work, we want to explore the nature and function of alpha sources in the auditory cortex. What are the neural sources of alpha activity, its response profile during auditory stimulation and its relationship with the evoked neural sources processing the sensory input.

[...]

In this work we performed SEEG recordings from the human auditory cortex to track the activity of the neural sources during two conditions: rest and pure tone stimulation. With ICA, we identified the main sources of alpha oscillations at rest (“oscillatory sources”) and those with a significant auditory evoked potential (AEP, “evoked sources”). First, we compared whether the sources responding to the stimulus were also those with highest alpha at rest (i.e., whether they were the same neural source or not). Second, we compared the power spectrum across both conditions to characterize sustained changes of alpha power. Finally, we analyzed the time-frequency response of the sources during pure-tone stimulation for a fine-grained exploration of stimulus induced alpha modulations.”

3) What about lower frequent oscillations given the relatively rhythmic auditory stimulation?

The spectral analysis does not seem well suited to extract peaks in lower frequency regions due to the 1/f property of the spectrum (in FOOOF the lower frequency cutoff strongly determines whether low frequent peaks are observed or not).

Unfortunately, the interval between stimuli (~1s) was too short to explore delta activities. Therefore, we measured the FOOOF between 2 and 30 Hz, to limit the confound effects from this rhythmicity. While these low-frequency oscillations are quite important for auditory processes, specially in speech, they are beyond the goals of our study and the possibilities of our task. We have included a paragraph in the discussion about this limitation (p-31):

“While we focus our work on alpha oscillations, these are not the only rhythms in the auditory cortex (Mai et al., 2016). For instance, theta (4-8 Hz) and delta (0-4 Hz) oscillations are key dynamics for speech processing, with the former tracking the syllabic time scale (Luo and Poeppel, 2007) and the latter associated with prosody (Inbar et al., 2023), pauses (Chalas et al., 2023) and other top-down (Fontolan et al., 2014) or linguistic processes (Chalas et al., 2024). Given the nature of our stimulus (short pure tones) and the short intertrial interval (~1s), these oscillations are beyond the scope and possibilities of this study. Further work should investigate the interaction between alpha and other brain oscillations, and particularly explore whether, in continuous stimulation contexts such as speech or music perception, the gating of information depends on contextual or predictive features.”

and also the removal of the average response might have removed phase-locked oscillations (also see point on evoked / induced below).

We have now included a comparison of the time-frequency map and inter-trial phase clustering (ITPC) with both with and without the averaged response (see below, answer to question 5).

4) The use of ICA to select components of interest is put forward as an important strength of the work (l. 80-84), or even a goal, but I was missing a clear link to the scientific goals / hypotheses. It is per se not so novel to use ICA for selection of components of interest rather than rejection of artifacts (see e.g. Chaumon & Busch 2014). Please clarify why the ICA has particular benefits for sEEG and why it can "remove the electrical reference and retrieve remote sources not sampled" (l. 80-84)?

The reviewer is correct in stating that ICA has been previously used to select components of interest (we have included Chaumon & Busch reference in the introduction). Nonetheless, this has been mostly done with EEG or MEG signals. We tried now to better clarify that the originality of our approach is its use on intracerebral EEG data. We used it to separate the

different neural sources of activity, i.e., extracting the multiple alpha sources of the auditory cortex. This step is necessary when the neural sources are close or overlap in the same SEEG contact as traditional referential and bipolar montages would not be able to separate the sources. This allowed us to differentiate the induced changes in alpha power (internal to each source) and the main sustained effect, particular of the sources with high alpha power oscillations but no evoked response.

We have better explained the use of ICA in the introduction (p.5):

“In intracerebral recordings, ICA has the potential to outperform traditional montages (Herreras et al., 2022, 2015). In referential montages, each contact records the activity of both local and remote sources that may be located far away (López-Madróna et al., 2023). One approach is to identify and remove the distant sources, whether it represents the electrical reference (Hu et al., 2007; Whitmore and Lin, 2016), or other neural sources (Michelmann et al., 2018). Moreover, rather than discarding this activity, it is possible to localize and analyze it, similar to the inverse problem in non-invasive recordings (López-Madróna et al., 2023; Medina Villalón et al., 2024). Bipolar montages are commonly used to measure local currents in a given location, but they may not recover the correct time-courses of local sources (Fernández-Ruiz and Herreras, 2013; Martín-Vázquez et al., 2013; Michelmann et al., 2018). For example, if two sources are located close to the same SEEG contact, the bipolar montage would not be able to separate them (López-Madróna et al., 2024). Therefore, ICA can be used to separate the multiple sources of alpha activity in the auditory cortex.”

5) I. 492: "Our work complements this result by describing two types of alpha suppression (sustained and induced). The evoked sources, mainly related to primary posteromedial structures, presented low alpha power during rest." No clear hypotheses were outlined concerning evoked or induced responses. Removing the average evoked response before computing time-frequency analyses also removes time-locked oscillations, which might occur due to phase resets in the stimulation condition, both in the alpha band and in lower frequency bands (delta).

The reviewer is right. We removed the AEP because its waveform was mainly composed by ~10 Hz activity, driving the time-frequency map for low frequencies. We have now run a new analysis and included a new figure (new Figure 5) comparing the time-frequency with and without AEP, as well as the effect in the ITPC. While most of the activity at low frequencies and between 0 and 200 ms was explained by the AEP, the effect in alpha suppression and high gamma activity remain unaltered.

However, as the frequencies of the AEP waveform overlapped those in the alpha range, we cannot dissociate which effects in the Time-Frequency plane and the ITPC are caused by the evoked activity or by the phase resetting of ongoing oscillations. Results p. 25:

“The time-frequency response of the evoked SEEG-ICs was also driven by the AEP (Figure 5). The waveform of the auditory response had a ~10 Hz pattern and most of the activity at low frequencies was strongly phase-locked with the stimulus (Figure 5a, right panel). The AEP also created a chimney effect in the time-frequency spectrum, close to stimulus onset. Therefore, neural activity in the entire frequency range was affected by the AEP and cannot be directly associated with changes in the oscillatory dynamics. To mitigate this effect, we also analyzed the response after removing the averaged evoked response (AEP; i.e., the phase-locked activity) from each trial (see Methods). Without the AEP contribution, the time-frequency map of the evoked sources had some similarities with the oscillatory sources (Figure 5b vs. Figure 3e). There was an early activation (here at very low frequencies; ~10 Hz) with a strong high-gamma response that started 25 ms after stimulus onset and lasted 200 ms, followed by a suppression at low frequencies (between 5 and 30 Hz), starting at 125 ms in these sources ($p < 0.05$, corrected with FDR). We repeated the analysis by averaging all the evoked SEEG-ICs per subject (number of observations equal to the number of patients with an evoked source), obtaining the same time-frequency pattern (Supplementary figure 3).

Figure 5: Time-frequency and phase locking responses of evoked SEEG-ICs:

- c) Averaged time-frequency response of evoked SEEG-ICs during pure tone stimulation (left panel). Framed areas represent clusters of significant modulation of activity compared to baseline (-300 -200 ms; $p < 0.01$, surrogate test). Inter-trial phase clustering (ITPC) representing the frequencies phase-locked with the stimulus (right panel)

d) Same analysis as in panel a but after removing the contribution of the averaged AEP to the evoked SEEG-IC time-courses. This process removes only the activity phase-locked with the stimulus.

To give more insights on the role of the phase, we have run a new analysis, proposed by reviewer 3, comparing the effect of instantaneous phase at stimulus onset with the amplitude of the evoked response, and gamma and alpha power. While none of the results were significant at the group level (Supplementary Figure 4), there was a tendency in the link between alpha phase and amplitude of AEP. Results p. 25:

“As the frequency of the AEP waveform overlapped with the alpha range, we could not dissociate whether the phase-locked activity was also contributed by a phase resetting of the ongoing oscillations. To better explore this scenario and knowing that the phase of ongoing oscillations in auditory cortex likely modulates its responsiveness to incoming stimuli (Ahveninen et al., 2024; Schroeder et al., 2008; Thézé et al., 2020), we measured how the instantaneous phase of alpha at stimulus onset influenced three different features of the response: amplitude of the AEP, high-gamma increase and alpha decrease (see Methods). We computed the weighted ITPC for each SEEG-IC, estimating their significance at the single level (permutation test). In only 2/60 oscillatory SEEG-ICs and 3/47 evoked SEEG-ICs, the phase of alpha oscillations influenced the intensity of the power response, either in the alpha or high-gamma range (Supplementary Figure 4). This indicates a marginal influence of alpha phase on the power response. For the amplitude of the AEP, 15/46 evoked sources did present a significant link with the instantaneous phase of alpha, although this effect was not significant at the group level.”

Supplementary Figure 4: Weighted ITPC between the instantaneous phase of alpha oscillations at stimulus arrival and three different features of the response: alpha (5-10 Hz) power, high-gamma (80-120 Hz) power and amplitude of the AEP. Each point represents one SEEG-IC. Dashed lines correspond to the significant threshold, established at 1.96. Values outside this range were considered as significant.

6) If I understood correctly, participants had their eyes closed during the resting state, but not during the auditory stimulation.

Previous work has shown that there might be different sources of alpha underlying these two states (Wöstmann et al. 2020), could this have an effect on divergence of alpha sources between conditions?

Participants did not have their eyes closed during both conditions and it is, indeed, a good practice to have the same state when comparing the effects. However, we expect the alpha effects due to eyes open/closed to be mainly related to occipital areas, as shown in Wöstmann et al. 2020. To reinforce our hypothesis, we have run a new experiment, only for reviewers, in one extra SEEG participant, comparing 5 minutes of resting with eyes open looking at a fixation point and 5 minutes of eyes closed. We used a bipolar montage and measured the power spectrum along all the contacts located in the auditory cortex (i.e., the same electrode as in our main work). The results have the same peak of activity centered at 7.5 Hz (see Review Figure 1) with no significant differences between both states.

Review Figure 1: Power spectrum across 12 bipolar contacts placed in the auditory cortex (mean \pm s.e.m.) of one patient. There is a clear peak of alpha activity around 7.5 Hz, with no significant differences between conditions.

7) Could you address in the discussion whether the fact that all participants are diagnosed with epilepsy might have any influences on the findings?

We have included more information related to SEEG implantation in the methods section as well as a new paragraph in the discussion, p. 31:

“Another general limitation inherent to every intracerebral study is the diagnosis of epilepsy. Although it cannot be fully addressed, several precautions can be taken to mitigate its impact. Every trace of epileptic activity was excluded from the analysis, and a partial or complete withdrawal of antiepileptic drug is done prior to the beginning of SEEG exploration, and none of the patients had their epileptogenic zone including the auditory areas. However, functional changes have been detected even in regions non-involved in the epileptic network (Lagarde et al., 2018). While they mainly affect the broadband

connectivity between regions (not studied in this work), we cannot completely rule out any effect in alpha oscillations.”

8) The methods lack some important details:

- AEP analyses: was a baseline applied before testing against 0?

Were the SEEG-ICs also z-scored for the analysis of evoked responses or just for computing power spectra?

Indeed, all the SEEG-ICs were z-scored before the analyses.

No baseline correction was applied during the AEP analysis, as the data was already high-pass filtered (0.16 Hz) and z-scored to remove very low frequency trends.

We have now corrected this information in the methods section.

- l.190/191: 'This averaged AEP was then fitted for each trial using a linear regression minimizing the difference between the single trial and the fitted AEP. Finally, we subtracted the fitted AEP 191 independently of each trial.'

I did not understand this part and how it produces a single trial signal to subtract from single trials. Please clarify.

We remove all the information explained by the averaged AEP at the single trial level. To do so, we subtracted, for each trial, the averaged AEP multiplied by a factor k that minimized the difference. The factor k should indicate the amplitude of the AEP at each trial. We used the linear regression to find this factor k.

We now explain this procedure in the methods section, p. 10:

“To limit the contribution of the AEPs during pure tone stimulation to the power spectrum, we also analyzed the time-courses after removing the averaged response from each trial (Pure Tone no AEP condition). First, we computed the AEP for each SEEG-IC. This averaged AEP was then fitted for each trial using a linear regression minimizing the difference between the single trial and the fitted AEP and it was subtracted for each trial. In other words, we subtracted to each trial the averaged AEP multiplied by a factor ‘k’ that minimized the result. The remaining time-course should contain all the activity that is not explained by the evoked response.”

- l 197: "We visually inspected all the SEEG-ICs to remove the components related to the reference or to remote sources, i.e., with a similar activity along all the contacts of the electrode (Figure 1d, 198 gray component). Then, we classified the remaining components as “oscillatory” sources if they presented a high oscillatory activity during resting state, “evoked” sources if they had a significant 200 AEP, both “oscillatory” and “evoked” sources, or “nonrelated”

components."

What does it mean for a component to present 'high oscillatory activity', and were there any measures taken to prevent biases by the raters?

We referred to the components with a significant peak in the power spectrum obtained by the foof analysis. We now explain this procedure in the Results section p. 11.

We then selected the 25% SEEG-ICs with highest power (see our answer below).

- Please provide the parameters used in the FOOF analysis.

We have included this information in the methods section, p.10:

"We selected the range from 2 to 30 Hz for the foof fit, a minimum bandwidth for peak detection of 0.5 Hz (twice the frequency resolution) and a minimum amplitude of twice the standard deviation of the aperiodic-removed power spectrum (see Supplementary Figure 1 for a distribution of all the identified peaks across SEEG-ICs and conditions). The knee parameter was fixed at zero."

- In the following the authors state that the highest significant peak was selected and then the 25% ICs with highest power. The rationale for choosing 25% sounds a bit handwavy, and might bias the results, as there could be a priori more or less oscillatory compared to evoked components.

Even though a control analysis was performed with different percentages, it seems to be a strong assumption to a-priori limit the number of oscillatory components. This might obscure oscillations in different frequency bands. As indicated above, it was not clear to me whether the approach was to solely focus on alpha oscillatory components, in which case the analyses could be more tailored towards this frequency band and its variations.

We apologize for the lack of clarity in our selection criteria. It was directly related to our scientific question, although it was not clearly stated in the text. Our goal was to test whether the neural sources processing the stimulus (i.e., those with a significant evoked response) were also the sources with the main alpha oscillations in rest, or if they were independent. The "evoked sources" were selected statistically. Then, to do a fair comparison, we selected a threshold to have the same number of "oscillatory sources" as "evoked sources", so we can check whether both groups overlap, or not. These were our main criteria.

It is true that, once we demonstrated that they were relatively independent, the 25% criterion is less relevant, as they don't need to be related to the number of evoked sources. Thus, we did a further control with different percentages to prove that our original criterion was not biasing the main results (Results p. 21).

We have better explained our choice in the main text, both in the Methods section p. 11:

“From all the SEEG-ICs, we chose the 25% with highest power (71 SEEG-ICs; 0.75 quantile), labeling them as oscillatory. As one of the goals of this study was to test whether the sources processing the input (i.e., evoked sources) were also the sources with the main alpha oscillatory activity in rest, we fixed this value to include SEEG-ICs from most of the electrodes in the analysis (21 out of 23 electrodes had an oscillatory source) while keeping a similar number of oscillatory and evoked source (71 versus 58, respectively).”

- I.254/255: It sounds important to first test whether the location is different, before addressing the standard deviation? The standard deviation could be biased by the location of the contact, for instance if it is close to the edge of the electrode.

Indeed we did test whether the location was different, but it was not clear in the text. We have rephrased it. (Results p. 18):

“The location of the SEEG-ICs differed for both types of sources (average of absolute lateral-medial axis locations in MNI space \pm s.d.: 50.39 ± 11.65 and 43.49 ± 8.89 for oscillatory and evoked SEEG-ICs; $p=0.0024$, t-test, $t=3.09$, $df=125$; Figure 2a), with evoked sources located in medial areas while oscillatory sources were more lateral. The contacts with maximal contribution of the oscillatory SEEG-ICs were distributed along the lateral-medial axis, in contrast to the evoked sources, which were clustered in more medial areas (Figure 2b). The oscillatory sources occupied a larger area (i.e., they were more distributed) than the evoked sources, with their location presenting a higher standard deviation across SEEG-ICs (s.d. of the absolute lateral-medial axis locations in MNI space: 11.65 and 8.89 for oscillatory and evoked SEEG-ICs, $p=0.03$, permutation test).”

References:

Chaumon M, Busch NA. Prestimulus neural oscillations inhibit visual perception via modulation of response gain. J Cogn Neurosci. 2014 Nov;26(11):2514-29. doi: 10.1162/jocn_a_00653

Wöstmann, M., Schmitt, L. M., & Obleser, J. (2020). Does closing the eyes enhance auditory attention? Eye closure increases attentional alpha-power modulation but not listening performance. Journal of cognitive neuroscience, 32(2), 212-225.

Reviewer #3 (Remarks to the Author):

In this paper, the authors use intracranial EEG and independent component analysis to describe two dissociable types of activity in the superior temporal lobe, one that oscillates at rest and does not display low-frequency evoked responses to sounds, and another with the opposite characteristics. The methods are appropriate and well-described, the results are described and illustrated eloquently, and the discussion is interesting and very reasonable. Overall, this paper represents a focused, important contribution to our understanding of how the brain's activity organizes at rest and changes during the processing of sensory inputs. I only have a few suggestions for the authors to entertain.

We appreciate the reviewer's positive evaluation of our work.

Major comments:

Re: the existence of two kinds of alpha suppression (induced and sustained): I would like to suggest that the authors explore the phase consistency of both types of alpha activity with respect to the auditory stimulus. An absence of inter-trial phase consistency of sustained alpha drop would be an additional suggestion that something non-specific is at play.

We have included a new analysis of inter-trial phase coherence (ITPC) to measure the phase consistency (new figure 5). Note that in our previous analysis we removed the AEP from the data to quantify the power, i.e., all the phase-locked activity (Figure 5b), so the alpha drop was not related to the phase consistency.

Thanks to the new analysis (figure 5a) we see that the ITPC is maximal for the time (100 ms) and frequency (~10 Hz) of the AEP, creating a "chimney" effect up to >80 Hz. This translates into an increase of power at ~10 Hz which is caused by the AEP rather than an oscillatory activity. Indeed, all the high-gamma activity and the posterior decrease of alpha activity remains unchanged after removal of the AEP, indicating that these effects were not phase-locked (evoked). Results p. 25:

"The time-frequency response of the evoked SEEG-ICs was also driven by the AEP (Figure 5). The waveform of the auditory response had a ~10 Hz pattern and most of the activity at low frequencies was strongly phase-locked with the stimulus (Figure 5a, right panel). The AEP also created a chimney effect in the time-frequency spectrum, close to stimulus onset. Therefore, neural activity in the entire frequency range was affected by the AEP and cannot be directly associated with changes in the oscillatory dynamics. To mitigate this effect, we also analyzed the response after removing the averaged evoked response (AEP; i.e., the phase-locked activity) from each trial (see Methods). Without the AEP contribution, the time-frequency map of the evoked sources had some similarities with the oscillatory sources (Figure 5b vs. Figure 3e). There was an early activation (here at very low

frequencies; ~ 10 Hz) with a strong high-gamma response that started 25 ms after stimulus onset and lasted 200 ms, followed by a suppression at low frequencies (between 5 and 30 Hz), starting at 125 ms in these sources ($p < 0.05$, corrected with FDR). We repeated the analysis by averaging all the evoked SEEG-ICs per subject (number of observations equal to the number of patients with an evoked source), obtaining the same time-frequency pattern (Supplementary figure 3).

Figure 5: Time-frequency and phase locking responses of evoked SEEG-ICs:

- e) Averaged time-frequency response of evoked SEEG-ICs during pure tone stimulation (left panel). Framed areas represent clusters of significant modulation of activity compared to baseline (-300 -200 ms; $p < 0.01$, surrogate test). Inter-trial phase clustering (ITPC) representing the frequencies phase-locked with the stimulus (right panel)
- f) Same analysis as in panel a but after removing the contribution of the averaged AEP to the evoked SEEG-IC time-courses. This process removes only the activity phase-locked with the stimulus.

Similarly, I would be curious to know about the impact of instantaneous phase at stimulus arrival on all types of stimulus-locked responses (low-frequency evoked potentials and broadband high-frequency power changes). The phase of ongoing oscillations in auditory cortex likely modulates its responsiveness to incoming stimuli (see e.g. Schroeder et al., TICS 2008; Thézé et al., Sci Adv 2020; and Ahveninen et al., J Neurosci 2024, in the multisensory context).

We have performed a new analysis of weighted ITPC, investigating whether the instantaneous phase of alpha (~ 8 Hz) correlated with the increase of gamma power, decrease of alpha power, or the amplitude of the evoked potential (Methods p. and supplementary figure 4):

Supplementary Figure 4: Weighted ITPC between the instantaneous phase of alpha oscillations at stimulus arrival and three different features of the response: alpha (5-10 Hz) power, high-gamma (80-120 Hz) power and amplitude of the AEP. Each point represents one SEEG-IC. Dashed lines correspond to the significant threshold, established at 1.96. Values outside this range were considered as significant.

While none of them were significant at the group level, the alpha phase modulated the amplitude of the AEP in 14 out of 47 SEEG-ICs (~30%). We now state in the Results section, p.25:

“[...] and knowing that the phase of ongoing oscillations in auditory cortex likely modulates its responsiveness to incoming stimuli (Ahveninen et al., 2024; Schroeder et al., 2008; Thézé et al., 2020), we measured how the instantaneous phase of alpha at stimulus arrival influenced three different features of the response: amplitude of the AEP, high-gamma increase and alpha decrease (see Methods). We computed the weighted ITPC for each SEEG-IC, computing the significance at the single level (permutation test). In only 2 out of 60 oscillatory SEEG-ICs and 3 out of 47 evoked SEEG-ICs, the phase of alpha influenced the intensity of the power response (Supplementary Figure 4). For the amplitude of the AEP, 15 out of 46 evoked sources did present a significant link with the instantaneous phase of alpha although it was not significant at the group level.”

Re: high-frequency power changes at both oscillatory vs. evoked sites: power is clearly lower in oscillatory sites, but what about onset latency? Finding a latency difference between the two types of response might help establish a processing hierarchy.

We have performed the new analysis (Figure 6b and 6c) and, as expected by the reviewer, there is a processing hierarchy, with high-gamma activity being earlier in evoked sites (more primary) compared to oscillatory sites. Indeed, this is another indication that there is a processing hierarchy, with an initial activation of the main components processing the stimulus in primary

areas (i.e., the sources with a significant AEP) followed by a second activation of the oscillatory sources, probably in charge in higher level cognitive functions and controlling the inhibition state of the network (Results p 27):

“Finally, we tested whether the activation time between oscillatory and evoked sources differ during pure tone stimulation. We measured the evoked-related power at high-gamma (80-120 Hz) frequencies (Figure 6b) and measured the onset latency of this activity. Evoked sources responded significantly faster than oscillatory sources (68 ± 41 ms and 33 ± 24 ms for oscillatory and evoked SEEG-ICs, respectively; t-test, $p < 0.0001$, $t = 4.71$, $df = 77$, $CI = 20, 50$).”

Figure 6: Comparison of the time-frequency dynamics between oscillatory and evoked SEEG-ICs

- a) Contrast between the time-frequency responses of oscillatory (Figure 3e) minus evoked (Figure 5b) sources. Delineated areas represent the clusters with significant difference ($p < 0.05$ corrected with FDR).
- b) Averaged gamma power during pure tone stimulation of oscillatory and evoked SEEG-ICs (mean \pm s.e.m.).
- c) Onset latency of gamma activity for each source type. Evoked sources had a significant early activation than oscillatory SEEG-ICs suggesting a processing hierarchy.

Minor points:

General note: would the authors consider replacing "state" (as in resting state) by "task" or "condition", in order to separate their two conditions? To me, "state" evokes varying states of vigilance (wakefulness vs. drowsiness, sleep, anesthesia, coma) rather than the particulars of what an awake and behaving person is asked to do (even when they are asked to not do anything in particular).

We have modified all the references to resting state along the text.

Introduction:

A quick comment re: resting iEEG activities: in addition to the Frauscher et al., 2018 reference, you might consider consulting Groppe et al., Neuroimage 2013, who basically did the same thing earlier.

We have included the new reference, were they also identified that the main activity was at 6-8 Hz.

Methods:

Description of the selected SEEG electrodes (Methods > SEEG recordings and Figure 1a): was the selection of patients and electrode shafts driven by the presence of one or more contacts in a given parcellation of the Brainnetome atlas, or else done by hand (visual inspection)? What were the regions targeted by the electrodes that passed through Heschl's gyrus or the planum temporale? From Figure 1a, I have the impression that the deep-most contact is in the thalamus, which is not a typical target of invasive EEG recordings for epilepsy surgery planning.

Actually, the thalamus was recorded in several of our participants and it is a main target at the Hospital La Timone (France). With a single electrode, it is possible to cover the auditory cortex and also reach the thalamus with the tip. We have further explained the implantation strategy in the methods section:

“There is no “standard” approach for the electrode implantation, as it is entirely guided by the hypotheses regarding the anatomical location of the epileptogenic zone (EZ). The goal is to identify the specific area for subsequent cortectomy. These hypotheses about the potential location of the EZ are formed based on non-invasive pre-surgical assessments (Phase I), which include a detailed clinical history, surface video-electroencephalographic (EEG) recordings, MRI, and 18FDG-PET scans. Consequently, electrode placement is tailored to each patient’s unique clinical profile rather than being standardized.

One of the most commonly explored areas is the perisylvian region, particularly when there is a need to determine whether the patient's epilepsy is temporal, temporo-perisylvian, or purely perisylvian. Misdiagnosing perisylvian epilepsy is a leading cause of surgical failure in temporal epilepsy. Perisylvian epilepsy can be located in various regions, including the insular cortex, frontal operculum, parietal operculum, temporal operculum, and the superior temporal and supramarginal gyri. In this region, electrodes are typically implanted orthogonally to the cortical surface to capture recordings along the electrode from areas like the posterior insula, the tip of Heschl's gyrus, and the planum temporale. Another approach involves implanting electrodes more anteriorly to target the superior temporal gyrus and ventral insula. The perisylvian region is also explored under the hypothesis of epilepsy originating from the inferior parietal lobule, pericentral area, or ventral prefrontal/premotor cortex.

The transverse gyrus (Heschl's gyrus), which includes the auditory cortex, plays a crucial role due to its connections with lower central regions and the inferior frontal gyrus. This region serves as a pathway through which seizures from the temporal pole and the anterior superior temporal gyrus can spread.”

Listening task (Methods > Experimental paradigm): was the listening done with eyes closed, like the resting state? If not, could this introduce bias?

Participants did not have their eyes closed during both conditions and it is, indeed, a good practice to have the same state when comparing the effects. However, we expect the alpha effects due to eyes open/closed to be mainly related to occipital areas, as shown in Wöstmann et al. 2020. To reinforce our hypothesis, we have run a new experiment, only for reviewers, in one extra SEEG participant, comparing 5 minutes of resting with eyes open looking at a fixation point and 5 minutes of eyes closed. We used a bipolar montage and measured the power spectrum along all the contacts located in the auditory cortex (i.e., the same electrode as in our main work). The results have the same peak of activity centered at 7.5 Hz (see Review Figure 1) with no big differences between both states.

Review Figure 1: *Power spectrum across 12 bipolar contacts placed in the auditory cortex (mean \pm s.e.m.) of one patient. There is a clear peak of alpha activity around 7.5 Hz, with no clear differences between conditions.*

Regarding the x-axis (Methods > Statistical analysis): would it be easier for the reader if the text referred to the lateral-medial axis?

Following the reviewer's advice, we have replaced x-axis by lateral-medial axis.

Regarding the permutation test on the variance in the x-axis location of oscillatory vs. evoked ICs (Methods > Statistical analysis): a small thing really, but when using permutation testing to compute the probability that a variable be observed by chance under the null hypothesis, the observed value of the variable must be added to the set of simulated values. The computation of the p-value is then the sum of the N simulated values and the observed one that are larger _or

equal_ to the observed value, divided by the number of simulated values + 1. That way, the smallest possible p-value is not 0/N, which is obviously problematic, but 1/(N+1). See e.g. the book by Manly (1997), or Phipson and Smyth, Stat Appl Genet Mol Biol 2010 (<https://doi.org/10.2202/1544-6115.1585>).

We agree with the reviewer and we have included his advice in the text. The new time-frequency figures have been computed adding the observed value to the distribution, and the results remain the same.

Results:

Example recordings (Figure 1b,c): given the high SNR of iEEG, would it be possible to show a single-trial AEP on panel c, rather than an average? It would allow the reader to grasp how oscillatory activity (depicted on panel b) attenuates during stimulus processing.

Although we have repeated the analysis, we have not found a good representative trial to replace the figure. Despite the reduction of oscillatory power, it is still quite high, and there are also other sources of activity (note that in panel C the y axis are 4 times lower than in panel B). Nevertheless, we have included a single trial in panel F (ICA).

Figure 1: Separation of oscillatory and evoked sources with ICA:

- Cerebral MRI scan (3D T1-weighted) – cross section with reconstruction of the SEEG electrode for patient 9. The location of each contact is represented with white rectangles.
- Example of monopolar recordings during rest. High amplitude oscillations can be appreciated in the superior channels (H'9 and H'10).
- Averaged AEP during presentation of pure tones at each contact. The highest response in amplitude is observed in channel H'4.
- Spatial profile of three SEEG-ICs across the electrode, representing their contribution to each contact. The purple and blue components have clear peaks in the profile, suggesting a local origin of the sources around these contacts. The grey component contributed almost equally to all the contacts, and hence reflect a remote source.
- SEEG-IC traces during the same time period as panel b. The oscillations visible in the raw SEEG were captured by the purple component, which was labeled as oscillatory source.
- Averaged AEP of each SEEG-IC (solid lines) and a single trial (dashed lines). Only the blue component has a significant response and was labeled as evoked source.

Location of ICs (Figure 2b): I suggest adding orientation labels to the views of the temporal lobes, indicating where the pole, planum, medial and lateral sides are.

We have added orientation labels to the plot.

Discussion:

I would like to suggest that the authors briefly explore how to test the hypothesis that they lay out in the last paragraph of the discussion (earlier auditory cortex being always "on" vs. the activity of higher-order areas being gated by resting oscillations). With continuous stimuli? or a more complex sequence of auditory stimuli that are more or less predictable of what comes next?

We have now added the following paragraph in the Discussion section, on p. 31:

“While we focus our work on alpha oscillations, these are not the only rhythms in the auditory cortex (Mai et al., 2016). For instance, theta (4-8 Hz) and delta (0-4 Hz) oscillations are key dynamics for speech processing, with the former tracking the syllabic time scale (Luo and Poeppel, 2007) and the latter associated with prosody (Inbar et al., 2023), pauses (Chalas et al., 2023) and other top-down (Fontolan et al., 2014) or linguistic processes (Chalas et al., 2024). Given the nature of our stimulus (short pure tones) and the short intertrial interval (~1s), these oscillations are beyond the scope and possibilities of this study. Further work should investigate the interaction between alpha and other brain oscillations, and particularly explore whether, in continuous stimulation contexts such as speech or music perception, the gating of information depends on contextual or predictive features.”

Pierre Mégevand

We would like to sincerely thank the reviewers for their thoughtful and positive assessment of our work. We greatly appreciate their time and constructive feedback, which has been invaluable in strengthening our manuscript.

Reviewer #3

No changes in power spectrum in temporal cortex when eyes are open and when they are closed: given that another reviewer also raised this question, and even though I understand that the authors will not want to add a figure with a single patient's data to their manuscript, they should still consider adding a sentence to the paper, stating that (for instance) based on their experience, no power change is expected in temporal cortex when eyes are open vs. close.

We have included a new paragraph in the discussion section (pag 16):

“A second limitation of our work is the effect of open vs. closed eyes on alpha activity. Both conditions are strongly linked in the occipital cortex (Klimesch, 1999). Based on our experience, this is not the case in the temporal cortex, which is more related to auditory rather than visual stimuli. However, a formal comparison of both conditions in the auditory cortex is necessary to accurately measure the impact of open vs. closed eyes.”

Targeting of S-EEG electrodes: I was not questioning the fact that superior temporal (incl. auditory) cortex is implanted en route to the insula, but rather the fact that the S-EEG electrode reaches all the way to the thalamus. As the authors say themselves, the goal of a S-EEG implantation is to plan a tailored cortical resection. The thalamus is never a target of resective epilepsy surgery. Additionally, the implantation of the thalamus is not mentioned in the French S-EEG guidelines (<https://www.sciencedirect.com/science/article/pii/S0987705317302873>). So, the implantation of a S-EEG electrode to the thalamus is not standard clinical practice, at least in most centers. This is what I think needs to be justified.

Indeed, the main target of the SEEG implantation was not the thalamus but the auditory cortex and the insula. Therefore, we removed the reference to the thalamus as main target in our last revision.

It should be noted that we did record the thalamus in some participants. While thalamic exploration via SEEG is relatively uncommon, these recordings play a critical role in guiding neuromodulation strategies and enhancing our understanding of the global organization of epileptic networks (Gadot et al., 2022; Carron et al., 2022). Our institution

has been conducting these recordings for over two decades. The SEEG electrode placement in the auditory cortex aligns with the pulvinar nucleus, enabling sampling of the thalamus without the need for additional electrodes. Moreover, it is also a safe procedure. We have not observed any hemorrhagic complications directly attributable to electrode insertion beyond the thalamus. To date, we have implanted over 2500 electrodes, with 250 specifically targeting the thalamus, over the past 12 years.

Gadot, R., Korst, G., Shofty, B., Gavvala, J. R., & Sheth, S. A. (2022). Thalamic stereoelectroencephalography in epilepsy surgery: a scoping literature review. Journal of Neurosurgery, 137(5), 1210-1225.

Carron, R., Pizzo, F., Trébuchon, A., & Bartolomei, F. (2022). Thalamic sEEG and epilepsy. Journal of Neurosurgery, 138(4), 1172-1173.